# FAIR SUBMODULAR COVER

**Wenjing Chen**[*]        **Shuo Xing**[*]        **Samson Zhou**[*]        **Victoria G. Crawford**[*]

## ABSTRACT

Machine learning algorithms are becoming increasing prevalent in the modern world, and as a result there has been significant recent study into algorithmic fairness in order to minimize the possibility of unintentional bias or discrimination in these algorithms. Submodular optimization problems also arise in many machine learning applications, including those such as data summarization and clustering where fairness is an important concern. In this paper, we initiate the study of the Fair Submodular Cover Problem (FSC). Given a ground set $U$, a monotone submodular function $f : 2^U \to \mathbb{R}_{\geq 0}$, and a threshold $\tau$, the goal of FSC is to find a balanced subset of $U$ with minimum cardinality such that $f(S) \geq \tau$. We first introduce discrete algorithms for FSC that achieve a bicriteria approximation ratio of $(\frac{1}{\varepsilon}, 1 - O(\varepsilon))$. We then present a continuous algorithm that achieves a $(\ln \frac{1}{\varepsilon}, 1 - O(\varepsilon))$-bicriteria approximation ratio, which matches the best approximation guarantee of submodular cover without a fairness constraint. Finally, we complement our theoretical results with a number of empirical evaluations that demonstrate the efficiency of our algorithms on instances of maximum coverage.

## 1    INTRODUCTION

From high-volume applications such as online advertising and smart devices to high-impact applications such as credit assessment, medical diagnosis, and self-driving vehicles, machine learning algorithms are increasingly prevalent in technologies and decision-making processes in the modern world. However, the amount of automated decision-making has resulted in concerns about the risk of unintentional bias or discrimination (Chouldechova, 2017; Kleinberg et al., 2018; Berk et al., 2021). For example, Chierichetti et al. (2017) noted that although machine learning algorithms may not be biased or unfair by design, they may nevertheless acquire and amplify biases already present in the training data available to the algorithms. Consequently, there has recently been significant focus on achieving algorithmic fairness for a number of fundamental problems, such as classification (Zafar et al., 2017), clustering (Chierichetti et al., 2017), data summarization (Celis et al., 2018b), and matchings (Chierichetti et al., 2019). Though various definitions have been proposed, there is no universal notion of fairness; indeed, determining the correct notion of fairness is an ongoing active line of research. In fact, Kleinberg et al. (2017) showed that three common desiderata of fairness (probabilistic calibration across classes, numerical balance across classes, and statistical parity) are inherently incompatible. Nevertheless, there has been significant focus recently (Chierichetti et al., 2017; Celis et al., 2018a;b;c; Chierichetti et al., 2019; El Halabi et al., 2020; Halabi et al., 2024) on the fairness notion that demands a solution to be balanced with respect to a sensitive attribute, such as ethnicity or gender.

In this work, we focus on fairness within submodular optimization. Submodular functions informally satisfy a diminishing returns property that is exhibited by many objective functions for fundamental optimization problems in machine learning. In particular, a function $f : 2^U \to \mathbb{R}$ is submodular if for every $X \subset Y \subset U$ and for every $x \in U \setminus Y$, we have $f(X \cup \{x\}) - f(X) \geq f(Y \cup \{x\}) - f(Y)$. We further assume $f$ is monotone, i.e., $f(Y) \geq f(X)$ for every $X \subset Y$. Thus, submodular optimization naturally arises in a wide range of applications, such as clustering and facility location (Gomes & Krause, 2010; Lindgren et al., 2016), document summarization (Lin & Bilmes, 2011; Wei et al., 2013; Sipos et al., 2012), image processing (Iyer & Bilmes, 2019), principal component analysis (Khanna et al., 2015), and recommendation systems (Leskovec et al., 2007; El-Arini & Guestrin, 2011; Chen & Crawford, 2024b; Mitrović et al., 2017; Yu et al., 2018;

---
[*]Texas A&M University, `jj9754@tamu.edu`, `shuoxing@tamu.edu`, `samsonzhou@gmail.com`, `vcrawford@tamu.edu`

Avdiukhin et al., 2019; Yaroslavtsev et al., 2020). While submodular maximization has received the most attention, submodular cover is also an important problem arising in machine learning applications (Iyer & Bilmes, 2013b; Mirzasoleiman et al., 2015; Soma & Yoshida, 2015; Norouzi-Fard et al., 2016; Mirzasoleiman et al., 2016; Ghuge et al., 2021; Ran et al., 2022; Chen & Crawford, 2024a). In monotone submodular cover (SC), the monotone submodular function arises in the constraint: Given oracle access to a monotone submodular function $f : 2^U \to \mathbb{R}$ and a threshold $\tau$, the goal of SC is to identify a subset $S \subset U$ of minimum cardinality such that $f(S) \geq \tau$.

Many applications of submodular functions such as in clustering (Gomes & Krause, 2010; Lindgren et al., 2016) and data summarization (Lin & Bilmes, 2011; Wei et al., 2013; Sipos et al., 2012), are also applications where algorithmic fairness is important. Motivated by this, fair submodular maximization (FSM) has been considered under both a cardinality constraint and a fairness constraint (Celis et al., 2018a; Halabi et al., 2020). In the fair setting, we assume that each element in $U$ is associated with a color $c$ that denotes a protected attribute, which allows partitioning the ground set $U$ into disjoint groups $U_1, \ldots, U_N$. Then informally, the goal is to maximize the objective while selecting a representative number of elements from each color. Surprisingly, there has been no previous work studying fairness for the submodular cover problem, to the best of our knowledge. Thus in this work, we initiate the study of fairness for monotone submodular cover[1].

**Definition 1** (Fair Submodular Cover (FSC)). *Given input threshold $\tau$, and bounds on the proportion of the elements in each group $p_c$ and $q_c$, FSC is to find $argmin_{S \subseteq U} |S|$ such that $p_c|S| \leq |S \cap U_c| \leq q_c|S|$ for all $c \in [N]$ and $f(S) \geq \tau$.*

This definition of fairness incorporates multiple other existing notions of fairness, such as diversity rules (Biddle, 2017; Cohoon et al., 2013), statistical parity (Dwork et al., 2012), or proportional representation rules (Monroe, 1995; Brill et al., 2017). To guarantee the existence of feasible subsets, we assume that the inputs satisfy $\sum_{c \in [N]} p_c \leq 1$ and $\sum_{c \in [N]} q_c \geq 1$[2]. To further illustrate FSC, we describe a fair data summarization application. Let $U$ be a dataset that is split into disjoint subsets $U_1, ..., U_N$ such that each subset represents some attribute. The function $f$ is a monotone and submodular function that measures the information contained in a subset $X \subseteq U$, such as a submodular information measure (Iyer et al., 2021). The values of $p_c$ and $q_c$ for all $c \in [N]$, and $\tau$, are input by the user. Then FSC asks to find a minimum size summary that contains sufficient information ($f(S) \geq \tau$), while maintaining a balanced representation amongst the attributes (determined by $p_c$ and $q_c$).

**Our contributions.** In this paper, we propose bicriteria approximation algorithms for FSC. An $(\alpha, \beta)$-bicriteria approximation algorithm for FSC returns a solution set $X$ that satisfies $|X| \leq \alpha|OPT|$, $p_c|X| \leq |X \cap U_c| \leq q_c|X|$ for all $c \in [N]$, and $f(X) \geq \beta\tau$, where $OPT$ is an optimal solution to the instance of FSC. Notice that the solution of a bicriteria algorithm for FSC always satisfies the fairness constraint. However, the constraint on the function value ($f(X) \geq \tau$) might be violated by a factor of $\beta$ and therefore the solution is not necessarily feasible. But, if $\beta$ to close to 1, we can get a solution that is close to being feasible. We now describe the main contributions of the paper:

- In our first result, we take advantage of the dual relationship between FSM and FSC, and present two algorithms that convert bicriteria approximation algorithms for FSM into bicriteria approximation algorithms for FSC in Section 2. The first algorithm, `convert-fair`, is designed to convert discrete algorithms for FSM into ones for FSC. In particular, `convert-fair` takes a $(\gamma, \beta)$-bicriteria approximation algorithms for FSM and converts it into a $((1 + \alpha)\beta, \gamma)$-bicriteria approximation algorithm for FSC. Our sec-

---

[1]Notice that it is not necessarily obvious what values of input $\tau$, and $p_c, q_c$ result in an instance of FSC having a feasible solution. In particular, because $f$ is monotone it is easy to check whether there exists a set $X$ such $f(X) \geq \tau$ by simply testing whether $\tau \leq f(U)$, but whether there exists such a set that also satisfies the fairness constraint is unclear. We discuss the question of existence of a feasible solution further in Appendix B.4, and throughout the paper assume that values of input $\tau$, and $p_c, q_c$ are chosen such that there does exist a feasible solution to the instance.

[2]Notice that this assumption is necessary: If $\sum_{c \in [N]} p_c > 1$, then $\sum_{c \in [N]} p_c|S| > |S|$. However, by the definition of fairness constraint in FSC, we can get $p_c|S| \leq |S \cap U_c|$. It then follows that $\sum_{c \in [N]} p_c|S| \leq \sum_{c \in [N]} |S \cap U_c| = |S|$, which is a contradiction, implying there are no feasible sets. Similarly, if $\sum_{c \in [N]} q_c \geq 1$, we can also prove that no feasible sets satisfy the constraint.

ond conversion algorithm, `convert-continuous`, takes a continuous $(\gamma, \beta)$-bicriteria approximation algorithm and converts it into a $((1+\alpha)\beta, \frac{(1-3\varepsilon)\gamma - 2\varepsilon}{1+3\varepsilon+\frac{2\varepsilon}{\gamma}})$-bicriteria approximation algorithm for FSC.

Using our above result, we are now able to convert existing algorithms for FSM into algorithms for FSC. In particular, the algorithm Fair-Greedy in El Halabi et al. (2020) can be converted into bicriteria $(1+\alpha, 1/2)$ for FSC. However, a factor of $1/2$ on the feasibility constraint of $f(X) \geq \tau$ means that our solution is relatively far from being feasible. Motivated by this, our subsequent results focus on developing bicriteria algorithms for FSC that produce solutions arbitrarily close to being feasible. In particular:

- In our second result, we develop the first bicriteria approximation algorithms for FSC that find nearly feasible solutions. In particular, we propose three bicriteria algorithms for FSM that can be paired with our converting algorithms in order to find approximate solutions for FSC that are arbitrarily close to meeting the constraint $f(S) \geq \tau$ in FSC. The first two algorithms are the discrete algorithms `greedy-fairness-bi` and `threshold-fairness-bi`, which both achieve bicriteria approximation ratios of $(1 - O(\varepsilon), \frac{1}{\varepsilon})$, but the latter makes less queries to $f$ compared to the former. The third algorithm is a continuous one, `cont-thresh-greedy-bi`, which achieves an improved $(1 - O(\varepsilon), \ln\frac{1}{\varepsilon} + 1)$ bicriteria approximation ratio but requires more queries to $f$. The theoretical analysis of all these algorithms depends on Lemma 2, which is one of our central technical contributions and distinguishes our approach from the traditional submodular cover problem.

By leveraging `greedy-fairness-bi` and `threshold-fairness-bi` as subroutine algorithms for `convert-fair` and `cont-thresh-greedy-bi` as a subroutine for `convert-continuous`, we can obtain algorithms for FSC with a bicriteria approximation ratio of $((1+\alpha)\frac{1}{\varepsilon}, 1 - O(\varepsilon))$ and $((1+\alpha)(\ln(\frac{1}{\varepsilon}) + 1), 1 - O(\varepsilon))$ respectively. In contrast, using existing algorithms for FSM along with the converting algorithm does not yield solutions that are very close to being feasible for FSC. Finally:

- We perform an experimental comparison between our discrete algorithms for FSC and the standard greedy algorithm (which does not necessarily find a fair solution) on instances of fair maximum coverage in a graph and fair image summarization. We find that our algorithms find fair solutions while the standard greedy algorithm does not, but at a cost of returning solutions of higher cardinality.

## 1.1 RELATED WORK

Celis et al. (2018a) first gave a $(1 - 1/e)$-approximation algorithm for fair monotone submodular maximization under a cardinality constraint, which is tight given a known $(1 - 1/e)$ hardness of approximation even without fairness constraints (Nemhauser & Wolsey, 1978). This is accomplished by converting their instance of FSM into monotone submodular maximization with a specific type of matroid constraint called a fairness matroid, which we describe in more detail in Section 1.2, and then using existing algorithms for submodular maximization with a matroid constraint. The standard greedy algorithm achieves a $1/2$-approximation ratio for the submodular maximization with a matroid constraint (Fisher et al., 1978), and in addition there exists approximation algorithms using the multilinear extension that achieve a $1 - 1/e$ approximation guarantee (Calinescu et al., 2007; Badanidiyuru & Vondrák, 2014). Halabi et al. (2024) gave a $(1 - 1/e)$-approximation algorithm for fair monotone submodular maximization under general matroid constraints, though their algorithm only achieves the fairness constraints in expectation. Fair submodular optimization has also been under both cardinality and matroid constraints in the streaming setting (El Halabi et al., 2020; Halabi et al., 2024).

For the classical submodular cover problem without fairness constraints and integral valued $f$, the standard greedy algorithm, where the element of maximum marginal gain is selected one-by-one until $f$ has reached $\tau$, has been shown to have an approximation ratio of $O(\log\max_{e \in U} f(e))$ (Wolsey, 1982). To deal with real-valued $f$ (as in our case), a slight variant of the greedy where we stop at $(1 - \varepsilon)\tau$ instead of $\tau$ has been shown to be a $(\ln(1/\varepsilon), 1 - \varepsilon)$-bicriteria approximation algorithm (Krause et al., 2008). On the other hand, set cover is a special case of fair submodular cover (FSC),

and by the result of Feige (1998), it is not possible to achieve a $(1 - o(1)) \cdot \ln(n)$ approximation to FSC unless NP has "slightly superpolynomial time algorithms". Therefore, a large part of our motivation in considering bicriteria approximation guarantees for FSC is to develop constant factor approximation at the price of a small reduction to feasibility. Indeed, for a fixed $\varepsilon$, our algorithms achieve a constant factor approximation. If we set $\varepsilon = 1/n$, our continuous algorithm (Algorithm 2) for FSM with the converting algorithm (Algorithm 4) achieves a $((1 + \alpha) \ln(n) + 1), 1 - 7/n)$ bicriteria approximation guarantee for an instance of FSC, and so is very close to being feasible while having an approximation guarantee close to the lower bound.

Although submodular maximization has received relatively more attention than submodular cover, the problems have a dual relationship and thus, a natural approach for submodular cover is to convert existing algorithms for submodular maximization into ones for cover (Iyer & Bilmes, 2013a; Chen & Crawford, 2024a). In particular, Iyer & Bilmes (2013a) showed that given a $(\gamma, \beta)$-bicriteria approximation algorithm for submodular maximization with a cardinality constraint, one can produce a $((1+\alpha)\beta, \gamma)$-bicriteria approximation algorithm for submodular cover by making $\log_{1+\alpha}(n)$ guesses for $|OPT|$ in the instance of submodular cover, running the submodular maximization algorithm with the cardinality constraint set to each guess, and returning the smallest solution with $f$ value above $\gamma\tau$. However, this approach does not take into account the fairness constraints and cannot be used to convert algorithms for FSM into ones for FSC.

## 1.2 PRELIMINARIES

We now present preliminary definitions and notation that will be used throughout the paper. $OPT$ refers to the optimal solution of a submodular optimization problem. We use $[N]$ to denote the set $\{1, 2, ..., N\}$. The marginal gain of adding an element $s$ to the subset $S$ is denoted as $\Delta f(S, s)$. In addition, for any vector $\vec{v} = (v_1, v_2, ..., v_N)$, and any $k \in \mathbb{R}$, we define $k\vec{v} = (kv_1, kv_2, ..., kv_N)$, and we define $\lceil \vec{v} \rceil = (\lceil v_1 \rceil, \lceil v_2 \rceil, ..., \lceil v_N \rceil)$ and $\lfloor \vec{v} \rfloor = (\lfloor v_1 \rfloor, \lfloor v_2 \rfloor, ..., \lfloor v_N \rfloor)$.

We now define the related problem of fair submodular maximization (FSM) of a monotone submodular function $f$ (El Halabi et al., 2020), as the search problem of $\max\{f(S) : S \subseteq U, l_c \leq |S \cap U_c| \leq u_c, \forall c \in [N], |S| \leq k\}$ where $l_c$ and $u_c$ are the bound of cardinality within each small group. Without loss of generality, in this problem, it is assumed that $\sum_{c \in [N]} u_c \geq k$. This is because if $\sum_{c \in [N]} u_c < k$, then $|S| = \sum_{c \in [N]} |S \cap U_c| \leq \sum_{c \in [N]} u_c \leq k$. Therefore, the problem is equivalent to setting $k = \sum_{c \in [N]} u_c$. Since for the cover problem, the objective is to minimize the cardinality of the solution set which means $|S|$ is not fixed as it is in FSM, therefore we introduced the definition of fairness for FSC as a natural modification of the above problem where the fairness constraint is a proportion of the solution size as opposed to a fixed value.

The set of subsets satisfying fairness constraint above for FSM is not a matroid. However, it was proven by El Halabi et al. (2020) that we can convert an instance of FSM into an instance of submodular maximization problem with a matroid constraint; we state this result as Lemma 3 in Appendix A. This matroid constraint is called a fairness matroid, which we denote as $\mathcal{M}_{fair}(\mathcal{P}, \kappa, \vec{l}, \vec{u}) = \{S \subseteq U : |S \cap U_c| \leq u_c, \forall c \in [N], \sum_{c \in [N]} \max\{|S \cap U_c|, l_c\} \leq k\}$, where $\mathcal{P} = \{U_1, ..., U_N\}$ is the partition of the ground set $U$, $k$ is the total cardinality constraint, $\vec{l}, \vec{u} \in \mathbb{N}^N$ are the lower and upper bound vectors respectively. Below we propose the idea of a $\beta$-extension of a fairness matroid, which we will use in our bicriteria algorithms for FSM.

**Definition 2.** *For any $\beta \in \mathbb{N}_+$, we define the $\beta$-extension of the fairness matroid to be $\mathcal{M}_\beta = \mathcal{M}_{fair}(\mathcal{P}, \beta\kappa, \beta\vec{l}, \beta\vec{u}) = \{S \subseteq U : |S \cap U_c| \leq \beta u_c, \forall c \in [N], \sum_{c \in [N]} \max\{|S \cap U_c|, \beta l_c\} \leq \beta\kappa\}$.*

Our continuous algorithms will use the multilinear extension of $f$, defined as follows.

**Definition 3.** *For any submodular objective $f : 2^U \to \mathbb{R}_+$ with $|U| = n$, the multi-linear extension of $f$ is defined as $\boldsymbol{F}(\boldsymbol{x}) = \sum_{S \subseteq U} \prod_{i \in S} x_i \prod_{j \notin S} (1 - x_j) f(S)$ where $\boldsymbol{x} \in [0, 1]^n$, and $x_i$ is the $i$-th coordinate of $\boldsymbol{x}$. If we define $S(\boldsymbol{x})$ to be a random set that contains each element $i \in U$ with probability $x_i$, then by definition, we have that $\boldsymbol{F}(\boldsymbol{x}) = \mathbb{E}f(S(\boldsymbol{x}))$.*

We now present the definitions of discrete and continuous algorithms with an $(\alpha, \beta)$-bicriteria approximation ratio for FSM, which is defined to find $\arg\max_{S \in \mathcal{M}_{fair}(U, k, \vec{l}, \vec{u})} f(S)$.

**Definition 4.** *A discrete algorithm for FSM with an $(\alpha, \beta)$-bicriteria approximation ratio returns a solution X such that*

$$f(X) \geq \alpha f(OPT) \qquad \forall c \in [N], \ |X \cap U_c| \leq \beta u_c, \qquad \sum_{c \in [N]} \max\{|X \cap U_c|, \beta l_c\} \leq \beta k.$$

*Here OPT is the optimal solution of the problem FSM, i.e., $OPT = \arg\max_{S \in \mathcal{M}_{fair}(P, k, \vec{l}, \vec{u})} f(S)$.*

By this definition, we have that an algorithm satisfies a $(\alpha, \beta)$-bicriteria approximation ratio for FSM i.f.f the output set $S$ satisfies $f(S) \geq \alpha f(OPT)$ and that $S \in \mathcal{M}_\beta$.

**Definition 5.** *A continuous algorithm with $(\alpha, \beta)$-bicriteria approximation ratio for FSM returns a fractional solution $\boldsymbol{x}$ such that*

$$\boldsymbol{F}(\boldsymbol{x}) \geq \alpha f(OPT), \qquad \boldsymbol{x} \in \mathcal{P}(\mathcal{M}_\beta).$$

*Here OPT is the optimal solution of the problem FSM, i.e., $OPT = \arg\max_{S \in \mathcal{M}_{fair}(P, \kappa, \vec{l}, \vec{u})} f(S)$.*

*$\mathcal{M}_\beta$ is the $\beta$-extension of the fairness matroid $\mathcal{M}_{fair}(P, \kappa, \vec{l}, \vec{u})$, and $\mathcal{P}(\mathcal{M}_\beta)$ is the matroid polytope of $\mathcal{M}_\beta$. More specifically, $\mathcal{P}(\mathcal{M}_\beta) = conv\{\boldsymbol{1}_S : S \in \mathcal{M}_\beta\}$.*

## 2 CONVERSION ALGORITHMS FOR FSC

In this section, we introduce two algorithms that make use of the dual relationship between FSC and FSM, and convert bicriteria approximation algorithms for FSM into ones for FSC. The first algorithm, `convert-fair`, is designed to convert discrete algorithms for FSM into ones for FSC. In particular, `convert-fair` takes an $(\gamma, \beta)$-bicriteria approximation algorithms for FSM that runs in time $\mathcal{T}(n, \kappa)$ and converts it into a $((1 + \alpha)\beta, \gamma)$-bicriteria approximation algorithm for FSC that runs in time $O(\frac{\log(|OPT|)}{\log(\alpha+1)} \mathcal{T}(n, (1+\alpha)|OPT|))$. However, because of the matroid constraint, better approximation guarantees for FSM may be achieved by a continuous algorithm that produce a fractional solution. Motivated by this, our second converting algorithm, `convert-continuous`, takes a continuous $(\gamma, \beta)$-bicriteria approximation algorithm (where guarantees are with respect to the multilinear extension as described in Section 1.2), and converts it into a $((1 + \alpha)\beta, \frac{(1-3\varepsilon)\gamma - 2\varepsilon}{1 + 3\varepsilon + \frac{2\varepsilon}{\gamma}})$-bicriteria approximation algorithm for FSC. In the next section, we will develop corresponding bicriteria approximation algorithms for FSM that can be used along with the results in this section in order to produce approximately optimal solutions for FSC that are arbitrarily close to being feasible.

For both algorithms, it is required that the sets $U_c$ for $c \in [N]$ be sufficiently large so that our method of constructing a solution does not run out of elements to pick. In particular, we assume that the instance of FSC satisfies that $\sum_{c \in [N]} \min\{q_c, \frac{|U_c|}{\beta(1+\alpha)|OPT|}\} \geq 1$. Recall from the definition of FSC that it is already assumed $\sum_{c \in [N]} q_c \geq 1$, so this assumption is essentially requiring that there be enough elements within each set $U_c$ of the partition, relative to parameters $\alpha$ and $\beta$, to ensure that the rank of the $\beta$-extension of the fairness matroid $\mathcal{M}_\beta$ (see definition in Section 1.2) is $\beta\kappa$. For example, with the existing algorithm for FSM in El Halabi et al. (2020), $\beta$ is 1 and therefore the assumption is met if $|U_c| \geq q_c(1 + \alpha)|OPT|$ for all $c \in [N]$. For our algorithms `greedy-fairness-bi` and `cont-thresh-greedy-bi` in Section 3, $\beta$ would be $1/\epsilon$ and $\ln(1/\epsilon)$ respectively, and the assumption is met if $|U_c| \geq q_c(1 + \alpha)|OPT|/\epsilon$ and $|U_c| \geq q_c(1 + \alpha)\ln(1/\epsilon)|OPT|$ for all $c \in [N]$ respectively.

We first consider our algorithm `convert-fair` for converting discrete algorithms for FSM. Pseudocode for `convert-fair` is provided in Algorithm 1. `convert-fair` takes as input an instance of FSC, a $(\gamma, \beta)$-bicriteria approximation algorithm for FSM, and a parameter $\alpha > 0$. Each iteration of the while loop from Line 2 to Line 4 corresponds to a guess $\kappa$ on the size of the optimal solution to the instance of FSC. For each guess $\kappa$, we have an instance of FSM with budget $\kappa$, fairness vector of lower bound $\kappa\vec{p}$, fairness vector of upper bound $\kappa\vec{q}$. We then run the algorithm for FSM on this instance to get a set $S$. Notice that this algorithm will convert the matroid corresponding to the instance of FSM into its $\beta$-extension. In Lines 6 to 8, `convert-fair` adds additional elements so that the lower bounds are met for every one of the partitions. Next in Lines 10 to 12, `convert-fair` then adds elements until the size constraint $\beta\kappa$ is met, without breaking the fairness constraints. Finally, `convert-fair` checks if the set $S$ satisfies $f(S) \geq \gamma\tau$. If it does not, the guess of optimal solution size increases by a factor of $1 + \alpha$ and the process repeats itself.

---

**Algorithm 1** `convert-fair`

---

**Input**: An FSC instance with threshold $\tau$, fairness parameters $\vec{p}$, $\vec{q}$, partition of $U$ $\mathcal{P}$, a $(\gamma, \beta)$-bicriteria approximation algorithm for FSM, $\alpha > 0$
**Output**: $S \subseteq U$

1:   $\kappa \leftarrow (1 + \alpha), S \leftarrow \emptyset$.
2:   **while** $f(S) < \gamma\tau$ **do**
3:      $S \leftarrow$ Run $(\gamma, \beta)$-approximation algorithm for FSM with fairness matroid $\mathcal{M}_{fair}(\mathcal{P}, \kappa, \lfloor \vec{p}\kappa \rfloor, \lceil \vec{q}\kappa \rceil)$
4:      $\kappa \leftarrow \lceil (1 + \alpha)\kappa \rceil$
5:      //Rounding the solution
6:      **for** $c \in [N]$ **do**
7:         **if** $|S \cap U_c| < \beta \lfloor p_c\kappa \rfloor$ **then**
8:            Add new elements from $U_c/S$ to $S$ until $|S \cap U_c| \geq \beta \lfloor p_c\kappa \rfloor$
9:      **if** $|S| < \beta\kappa$ **then**
10:     **for** $c \in [N]$ **do**
11:        **while** $|S| < \beta\kappa$ and $|S \cap U_c| < \beta \lceil q_c\kappa \rceil$ **do**
12:           Add new elements in $U_c/S$ to $S$
13: **return** $S$

---

We now state the theoretical results for `convert-fair` below in Theorem 1. We defer the proof and analysis of the proof of Theorem 1 to Appendix C.1.

**Theorem 1.** *Suppose $\sum_{c \in [N]} \min\{q_c, \frac{|U_c|}{\beta(1+\alpha)|OPT|}\} \geq 1$. Then any $(\gamma, \beta)$-bicriteria approximation algorithm for FSM that returns a solution set in time $\mathcal{T}(n, \kappa)$ can be converted into an approximation algorithm for FSC that is a $((1 + \alpha)\beta, \gamma)$-bicriteria approximation algorithm that runs in time $O(\frac{\log(|OPT|)}{\log(\alpha+1)} \mathcal{T}(n, (1 + \alpha)|OPT|))$.*

We now present our algorithm `convert-continuous` for converting continuous algorithms. To motivate it, notice that here we can't directly use the converting theorem for discrete algorithms. There are two reasons for this: (i) The output solution is fractional so we need a rounding step, and (ii) the bicriteria approximation ratio for the continuous algorithms is on the value of the multi-linear extension and we don't have exact access to $F$, so we can't check directly if $\mathbf{F}(\mathbf{x}) \geq \gamma\tau$ as we did on Line 2 of `convert-fair`. Then we develop the converting algorithm `convert-continuous` for continuous algorithms. The key idea of the converting theorem is similar to `convert-fair`, so we defer the pseudocode of `convert-continuous` to Algorithm 4 in Section C.2 of the appendix. Here we describe the major differences. For each guess of optimal solution size $\kappa$, `convert-continuous` invokes the continuous subroutine algorithm for FSM to obtain a fractional solution $\mathbf{x}$. Since $\mathbf{F}$ can't be queried exactly in general, we estimate $\mathbf{F}(\mathbf{x})$ by taking a sufficient number of samples in Line 4. Once the estimate of $\mathbf{F}(\mathbf{x})$ is higher than $\gamma\tau$, we use the pipage rounding technique to convert $\mathbf{x}$ into a subset $S$, and then use the rounding procedure analogous to that in Lines 6 to 12 in Algorithm 1 to obtain a solution set with fairness guarantee. Notice that since the solution set obtained from the pipage rounding step is only guaranteed to satisfy that $\mathbb{E}f(S) \geq \mathbf{F}(\mathbf{x})$, the approximation guarantee on the function value in Theorem 5 holds in expectation. The corresponding theoretical guarantees for `convert-continuous` are stated below in Theorem 2. The proof of Theorem 2 can be found in Section C.2 of the appendix.

**Theorem 2.** *Suppose $\sum_{c \in [N]} \min\{q_c, \frac{|U_c|}{\beta(1+\alpha)|OPT|}\} \geq 1$. Then with probability at least $1 - \delta$, any $(\gamma, \beta)$-bicriteria approximation algorithm for FSM that returns a solution set in time $\mathcal{T}(n, \kappa)$ with probability at least $1 - \frac{\delta}{n}$ can be converted into an approximation algorithm for FSC that is a $((1 + \alpha)\beta, \frac{(1-3\varepsilon)\gamma - 2\varepsilon}{1+3\varepsilon + \frac{2\varepsilon}{\gamma}})$-bicriteria approximation algorithm where $\frac{(1-3\varepsilon)\gamma - 2\varepsilon}{1+3\varepsilon + \frac{2\varepsilon}{\gamma}}$ holds in expectation. The query complexity is at most $O\left(\frac{\log(|OPT|)}{\log(\alpha+1)} \mathcal{T}(n, (1 + \alpha)|OPT|) + \frac{n \log_{1+\alpha} |OPT|}{\varepsilon^2} \log \frac{n}{\delta}\right)$.*

## 3 BICRITERIA ALGORITHMS FOR FSM

In the last section, we propose converting algorithms to convert bicriteria algorithms for FSM into ones for FSC. Existing algorithms for FSM can be used as the input $(\gamma, \beta)$-bicriteria subroutine, but these algorithms all return feasible solutions to the instance of FSM and have guarantees of $\beta = 1$ and $\gamma \leq 1 - 1/e$. After applying the converting algorithms in Section 2 this results in solutions for FSC that are far from feasible. For example, the greedy algorithm for FSM proposed has $\gamma = 1/2$ and $\beta = 1$, and the continuous greedy algorithm for FSM has $\gamma = 1 - 1/e$ and $\beta = 1$ (Celis et al., 2018a).

In this section, we propose new algorithms that can be used for FSM where $\gamma$ is arbitrarily close to 1 and $\beta > 1$. As a result, the algorithms proposed in this section can be paired with the converting theorems in Section 2 to find solutions to our instance of FSC that are arbitrarily close to being feasible. We propose three bicriteria algorithms for FSM. The first two algorithms are the discrete algorithms `greedy-fairness-bi` and `threshold-fairness-bi`, which both achieve bicriteria approximation ratios of $(1 - O(\varepsilon), \frac{1}{\varepsilon})$ where the former is in time $O(n\kappa/\varepsilon)$ and the latter in time $O(\frac{n}{\varepsilon} \log \frac{\kappa}{\varepsilon})$. The third algorithm is a continuous one, `cont-thresh-greedy-bi`, which achieves a $(1 - O(\varepsilon), \ln \frac{1}{\varepsilon} + 1)$ bicriteria approximation ratio in time $O\left(\frac{n\kappa}{\varepsilon^4} \ln^2(n/\varepsilon)\right)$.

In the case of submodular maximization with a cardinality constraint without fairness, one can find a solution with $f$ value that is a factor of $1 - \varepsilon$ off of that of the optimal solution by greedily adding $O(\ln(1/\varepsilon))\kappa$ elements beyond the cardinality constraint $\kappa$. However, existing algorithms for FSM transform the instance into an instance of submodular maximization subject to a fairness matroid constraint, and it is not clear how one can take an analogous approach and produce an infeasible solution in order to get a better approximation guarantee when dealing with a matroid constraint while maintaining a fair solution. We propose the $\beta$-extension of a fairness matroid, defined in Section 1.2, in order to get a $(\gamma, \beta)$-bicriteria algorithm for FSM with $\gamma > 1 - 1/e$. In particular, we will return a solution that is a feasible solution to the $\beta$-extension of the fairness matroid corresponding to our instance of FSM.

We now introduce two lemmas concerning the $\beta$-extension of a matroid that will be needed for our algorithms and their theoretical analysis. Before we proceed to present our algorithms, we first introduce the following general lemma that helps to build a connection between the fairness matroid $\mathcal{M}$ and its $\beta$-extension $\mathcal{M}_\beta$. For the sake of simplicity in notation throughout this section and its subsequent proofs, we use the notation $\mathcal{M}_\beta$, representing the $\beta$-extension of $\mathcal{M}_{fair}(P, \kappa, \vec{l}, \vec{u})$, which is defined in Section 1.2. Since the bicriteria algorithm for FSM is designed as a subroutine for the converting theorem with input $\mathcal{M}_{fair}(P, \kappa, \vec{p}\kappa, \vec{q}\kappa)$, we have that here $l_c = \lfloor p_c \kappa \rfloor$ and $u_c = \lceil q_c \kappa \rceil$. From the fact that $\sum_{c \in [N]} p_c \leq 1 \leq \sum_{c \in [N]} q_c$, we have that $\sum_{c \in [N]} l_c \leq \kappa \leq \sum_{c \in [N]} u_c$. Since the bicriteria algorithm for FSM is used as a subroutine for the converting theorems for FSC, where we assume $\sum_{c \in [N]} \min\{q_c, \frac{|U_c|}{\beta(1+\alpha)|OPT|)}\} \geq 1$ in both Theorem 1 and Theorem 2, we have that $rank(\mathcal{M}_{fair}(P, \kappa, \vec{l}, \vec{u})) = \kappa$ and that $rank(\mathcal{M}_\beta) = \beta\kappa$. Therefore, we have the following Lemma 1.

**Lemma 1.** $\sum_{c \in [N]} l_c \leq \kappa \leq \sum_{c \in [N]} u_c$ and $rank(\mathcal{M}_{fair}(P, \kappa, \vec{l}, \vec{u})) = \kappa$, $rank(\mathcal{M}_\beta) = \beta\kappa$.

Next, we present Lemma 2, which is one of the most interesting and novel parts of our analysis in proposing the bicriteria algorithm for FSM, and is necessary in the proof of all the bicriteria algorithm for FSM. Notice that in the bicriteria algorithm for submodular maximization with cardinality constraint, the greedy step is achieved by simply adding the element with the highest marginal gain. However, since we need to consider the fairness constraint in our paper, we can only add the element that makes the solution set feasible. To prove the theoretical guarantee, we have to construct a mapping from the solution set to the optimal solution. To tackle this difficulty, we propose and analyze Lemma 2, which guarantees the existence of such a mapping.

**Lemma 2.** For any $\beta \in \mathbb{N}_+$ and any fairness matroid $\mathcal{M}_{fair}(P, \kappa, \vec{l}, \vec{u})$, denote $\mathcal{M}_\beta$ as the $\beta$-extended fairness matroid of $\mathcal{M}_{fair}(P, \kappa, \vec{l}, \vec{u})$. Then for any set $S \in \mathcal{M}_\beta$ with $|S| = \beta\kappa$, $T \in \mathcal{M}_{fair}(P, \kappa, \vec{l}, \vec{u})$ with $|T| = \kappa$, and any permutation of $S = (s_1, s_2, ..., s_{\beta\kappa})$, there exist a sequence $E = (e_1, e_2, ..., e_{\beta\kappa})$ such that each element in $T$ appears $\beta$ times in $E$ and that

$$S_i \cup \{e_{i+1}\} \in \mathcal{M}_\beta, \qquad \forall i \in \{0, 1, ..., \beta\kappa\}$$

*where $S_i = (s_1, s_2, ..., s_i)$ and $S_0 = \emptyset$.*

Notice that this mapping is not straightforward and trivial due to two reasons. First of all, the size of the optimal solution and the solution set are different. Second, the portion of $|S \cap U_c|$ over $|T \cap U_c|$ can be different for different subroup $c \in [N]$. To prove Lemma 2, we construct the mapping by an iterative proof and by dealing with different cases for each step. This lemma is of great importance in our analysis as it guarantees the existence of a mapping from the solution set to the set containing $\beta$ copies of the $OPT$. The detailed proof of this lemma is deferred to the appendix.

On the other hand, we highlight that this lemma reveals an important and general fact about the fairness matroid: For each base set $T \in \mathcal{M}_{fair}(P, \kappa, \vec{l}, \vec{u})$, and each subset $S \in \mathcal{M}_\beta$ that is a base set, we can find a mapping from $T$ to a sequence $E$ that contains $\beta$ copies of $T$ such that $S_i \cup \{e_i\}$ is always feasible for $\mathcal{M}_\beta$. Notice that since $\mathcal{M}_\beta$ is a matroid, then for any subset $S_1 \subseteq S_2$, if $S_2 \in \mathcal{M}_\beta$ then $S_1 \in \mathcal{M}_\beta$. Consequently, the above lemma holds for not only just base set of $\mathcal{M}_\beta$, but also for any subset of $\mathcal{M}_\beta$ by adding dummy variables to the end of the sequence $S$ if the number of elements in $S$ is less than $\beta\kappa$. Building upon this lemma, we propose three bicriteria algorithms in the following part.

## 3.1 Discrete Bicriteria Algorithms for FSM

We now analyze two discrete bicriteria algorithms for FSM, `greedy-fair-bi` and `threshold-fairness-bi`. Let SMMC refer to the problem of monotone submodular maximization with a matroid constraint. `greedy-fair-bi` is based on the standard greedy algorithm which is well-known to produce a feasible solution with a $1/2$ approximation guarantee in $O(nk)$ time for SMMC (Fisher et al., 1978), where $k$ is the rank of the matroid. `greedy-fair-bi` proceeds in a series of rounds, where at each round we select the element $x \in U$ with the highest marginal gain to $f$ that stays on the $1/\varepsilon$-extension of the fairness matroid corresponding to the instance of FSM, i.e. $S \cup \{x\} \in \mathcal{M}_{1/\varepsilon}$. `threshold-fairness-bi` is based on the threshold greedy algorithm (Badanidiyuru & Vondrák, 2014), which is also a $1/2 - \varepsilon$ approximation for SMMC but requires only $O(n \log k)$ queries of $f$. `threshold-fairness-bi` iteratively makes passes through the universe $U$ and adds all elements into its solution with marginal gains exceeding $\tau$ that are feasible with respect to the $1/\varepsilon$-extension of the fairness matroid, and this threshold is decreased by $1 - \varepsilon$ at each round until it falls below a stopping criterion. Notice that these algorithms specifically use the $\beta$-extension of the fairness matroid, and therefore do not apply to the more general setting of submodular maximization with a matroid constraint. Pseudocode for the algorithms `greedy-fair-bi` and `threshold-fairness-bi` are included in Appendix D as Algorithms 5 and Algorithm 6 respectively.

We now present the theoretical guarantees of `greedy-fair-bi` and `threshold-fairness-bi`. The key benefit of these algorithms over existing ones for FSM is that by making $\varepsilon$ arbitrarily small and using `convert-fair` in Section 2, we have algorithms for FSC that are arbitrarily close to being feasible. In particular, if we use `greedy-fair-bi` as a subroutine in `convert-fair`, we have a $(\frac{1}{\varepsilon} + 1, 1 - \varepsilon)$-bicriteria algorithms for FSC in $\mathcal{O}(n \log(n)\kappa/\varepsilon^2)$ queries of $f$. If we use `threshold-fairness-bi`, we get a similar approximation guarantee in $\mathcal{O}(n \log(n) \log(\kappa/\varepsilon)/\varepsilon^2)$ queries of $f$. The proofs of both of these theorems can be found in Section D of the appendix.

**Theorem 3.** *Suppose that `greedy-fairness-bi` is run for an instance of FSM, then `greedy-fairness-bi` outputs a solution $S$ that satisfies a $(1 - \varepsilon, \frac{1}{\varepsilon})$-bicriteria approximation guarantee in at most $O(n\kappa/\varepsilon)$ queries of $f$.*

**Theorem 4.** *Suppose that `threshold-fairness-bi` is run for an instance of FSM with $\varepsilon \in (0, 1)$. Then `threshold-fairness-bi` outputs a solution $S$ that satisfies a $(1 - 2\varepsilon, \frac{1}{\varepsilon})$-bicriteria approximation guarantee in at most $O(n/\varepsilon \log(\kappa/\varepsilon))$ queries of $f$.*

## 3.2 Continuous Algorithms for FSM

A downside to the discrete greedy algorithms proposed in Section 3 is that we are above our budget $\kappa$ by a factor of $1/\varepsilon$, which is weaker than the analogous guarantee of $\ln(1/\varepsilon)$ that the greedy algorithm gives for submodular maximization with a cardinality constraint without fairness. We now introduce

---

**Algorithm 2** `cont-thresh-greedy-bi` (`cont-bi`)

---

1: **Input:** $\varepsilon, \delta, \mathcal{M} \in 2^U$
2: $\mathbf{x} \leftarrow \mathbf{0}$
3: $d := \max_{s \in \mathcal{M}} f(s)$,
4: **for** $t = 1$ to $1/\varepsilon$ **do**
5: $\quad B \leftarrow$ `decreasing-threshold-procedure` $(\mathbf{x}, \varepsilon, \delta, d, \mathcal{M})$
6: $\quad \mathbf{x} \leftarrow \mathbf{x} + \varepsilon \cdot \mathbf{1}_B$
7: **return** $\mathbf{x}$

---

**Algorithm 3** `decreasing-threshold-procedure` (`DTP`)

---

1: **Input:** $\mathbf{x}, \varepsilon, \delta, d, \mathcal{M} \in 2^U$
2: $B \leftarrow \emptyset$
3: Denote $\mathcal{M}_{fair}(P, \kappa \ln(1/\varepsilon), \vec{l} \ln(1/\varepsilon), \vec{u} \ln(1/\varepsilon))$ as $\mathcal{M}_{\ln(1/\varepsilon)}$.
4: **for** $w = d; w > \frac{\varepsilon d}{\kappa}; w = w(1 - \varepsilon)$ **do**
5: $\quad$ **for** $u \in U$ **do**
6: $\quad\quad X = \Delta f(S(\mathbf{x} + \varepsilon \mathbf{1}_B), u)$
7: $\quad\quad$ **if** $B \cup \{u\} \in \mathcal{M}_{\ln(1/\varepsilon)+1}$ **then**
8: $\quad\quad\quad \hat{X} \leftarrow$ average over $\frac{3\kappa}{\varepsilon^2} \log \frac{4n^4}{\varepsilon^3}$ samples from $\mathcal{D}_X$
9: $\quad\quad\quad$ **if** $\hat{X} \geq w$ **then**
10: $\quad\quad\quad\quad B \leftarrow B \cup \{u\}$
11: $\quad w = w(1 - \varepsilon)$
12: **return** $B$

---

and analyze our continuous algorithm `cont-thresh-greedy-bi` (`cont-bi`), which produces a fractional solution for FSM that achieves a $(1 - O(\varepsilon), \ln(1/\varepsilon) + 1)$-bicriteria approximation ratio in $O\left(n\kappa \ln^2(n)\right)$ time. `cont-bi` is based on the continuous threshold greedy algorithm of Badanidiyuru & Vondrák (2014). Compared to the discrete algorithms presented in Section 3.1, `cont-bi` improves the ratio on the cardinality of the solution from $O(1/\varepsilon)$ to $O(\ln(1/\varepsilon))$, and therefore has as strong of guarantees as the greedy algorithm without fairness. We can achieve a discrete solution with an arbitrarily small loss by employing rounding schemes, like swap rounding (Chekuri et al., 2010), on the returned fractional solution $\mathbf{x}$.

`cont-bi` iteratively takes a step of size $\varepsilon$ in the direction $\mathbf{1}_B$, where $\mathbf{1}_B$ is the indicator function of a set $B \subseteq U$, over $1/\varepsilon$ iterations. At each step, the set $B$ is determined by the subroutine `decreasing-threshold-procedure` (`DTP`). `DTP` builds $B$ over a series of rounds corresponding to thresholds $w$, where $w$ begins as the max singleton marginal gain and the rounds exit once $w$ is sufficiently small. During each round, we iterate over the universe $U$, and if an element $u \in U$ can be added to $B$ while staying on the $\ln(1/\varepsilon) + 1$-extension of the fairness matroid, then we approximate the multilinear extension $f$ and add $u$ to $B$ if and only if the marginal gain is above $w$. Pseudocode for `cont-bi` is provided in Algorithm 2, and pseudocode for `decreasing-threshold-procedure` is provided in Algorithm 3.

**Theorem 5.** *Suppose that Algorithm 2 is run for an instance of FSM, then with probability at least $1 - \frac{1}{n^2}$, `cont-thresh-greedy-bi` outputs a solution $S$ that satisfies a $(1 - 7\varepsilon, \ln(\frac{1}{\varepsilon}) + 1)$-bicriteria approximation guarantee in at most $O\left(\frac{n\kappa}{\varepsilon^4} \ln^2(n/\varepsilon)\right)$ queries of $f$.*

By applying a converting theorem, we can obtain the algorithm for submodular cover that achieves an approximation ratio of $((1 + \alpha)(\ln(\frac{1}{\varepsilon}) + 1), 1 - O(\varepsilon))$, which aligns with the best-known results for bicriteria submodular cover without the fairness constraint (Chen & Crawford, 2024a; Iyer & Bilmes, 2013b). The detailed theoretical guarantee and proof of the algorithm can be found in Corollary 5.1 in the appendix.

## 4 EXPERIMENTS

In this section, we evaluate several of our algorithms for FSC on instances of fair maximum coverage, where the objective is to identify a set of fixed nodes that optimally maximize coverage within

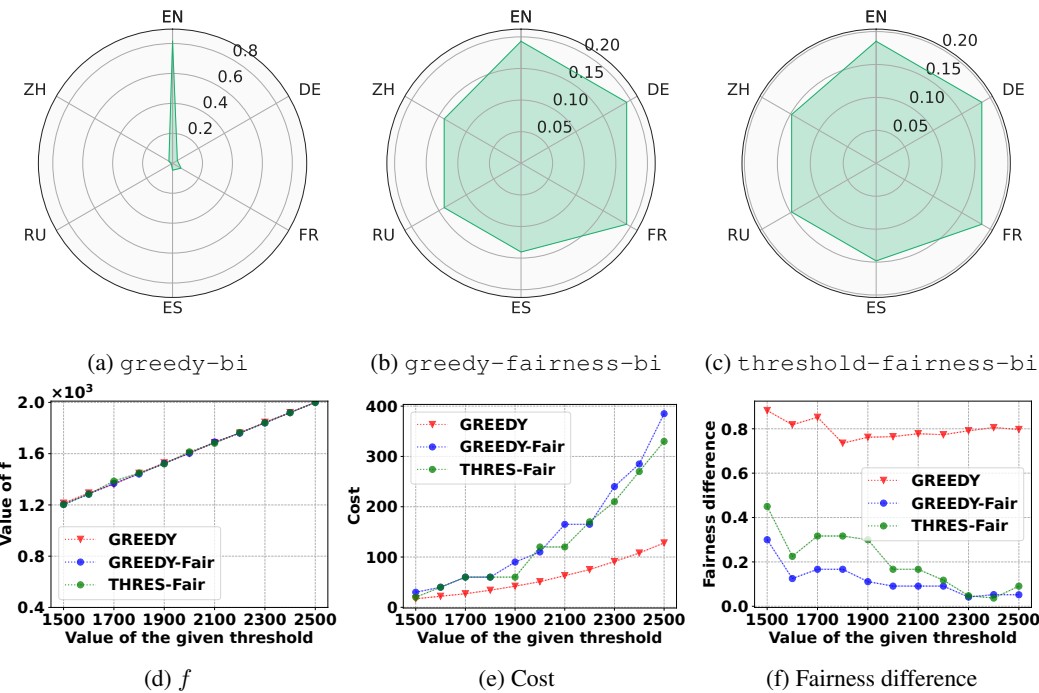

Figure 1: Performance comparison on the Twitch_5000 dataset for Maximum Coverage. 1a, 1b, 1c illustrate the distribution of users speaking different languages in the solutions produced by various algorithms with $\tau = 2400$. $f$: the value of the objective submodular function. Cost: the size of the returned solution. Fairness difference: $(\max_c |S \cap U_c| - \min_c |S \cap U_c|)/|S|$.

a graph. The dataset utilized is a subset of the Twitch Gamers dataset (Rozemberczki & Sarkar, 2021), comprising 5,000 vertices (users) who speak English, German, French, Spanish, Russian, or Chinese. We aim to develop a solution with a high $f$ value exceeding a given threshold $\tau$ while ensuring a fair balance between users who speak different languages. Additional discussion about the application as well as experimental setup including parameter settings are included in Appendix G. In addition, we include experiments on instances of fair image summarization in Appendix G.

We evaluate the performance of our discrete bicriteria algorithms `greedy-fair-bi` and `threshold-fairness-bi` for FSM as subroutines in our algorithm `convert-fair`. In addition, we consider the baseline algorithm, `greedy-bi`, which is the standard greedy algorithm for submodular cover without fairness. Figures 1a, 1b and 1c showcase the distribution of users speaking different languages in the solutions produced by these algorithms with $\tau = 2400$. Figures 1d, 1e and 1f present the performance of these algorithms ($f$ value, cost, and fairness difference) for varying values of $\tau$. As shown in Figure 1a, with $\tau = 2400$, over $80\%$ of the users in the solution returned by `greedy-bi` are English speakers, which indicates a lack of fairness in user language distribution. While the solutions produced by `greedy-fair-bi` and `threshold-fairness-bi` exhibit significantly fairer distributions across different languages, demonstrating the effectiveness of our proposed algorithms. Further, as the value of given $\tau$ increases, the magnitude of this difference also increases (see Figure 1f). Figure 1d showcases that for all these algorithms the objective function value $f(S)$ scales almost linearly with the threshold $\tau$, which aligns with the theoretical guarantees of the approximation ratio. Additionally, as shown in Figure 1e, the cost of the solutions returned by our proposed algorithms is higher than that of the solution from `greedy-bi`. This is an expected trade-off, as our algorithms have to include more elements to maintain the approximation ratio while ensuring fairness. Overall, our proposed algorithms are efficient and effective in producing a fair solution.

ACKNOWLEDGEMENTS

Samson Zhou is supported in part by NSF CCF-2335411. The work was conducted in part while Samson Zhou was visiting the Simons Institute for the Theory of Computing as part of the Sublinear Algorithms program. Victoria Crawford is supported in part by the Seed Program for AI, Computing, and Data Science created by the Texas A&M Institute for Data Science.

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

## A    OMITTED LEMMA OF SECTION 1.2

**Lemma 3** ((El Halabi et al., 2020))**.** *If $f$ is monotone, then solving FSM is equivalent to the problem below.*

$$\max_{S \in U} f(S)$$
$$s.t. \quad |S \cap U_c| \leq u_c \quad \forall c \in [N]$$
$$\sum_{c \in [N]} \max\{|S \cap U_c|, l_c\} \leq k$$

## B    ADDITIONAL DISCUSSION

### B.1    ADDITIONAL APPLICATIONS OF FSC

We first describe an additional number of applications of the fair submodular cover (FSC) problem.

**Data summarization.** The goal of data summarization is to find a small subset of a dataset, such as images, documents, etc, that summarizes the dataset. Monotone submodular functions have commonly been used in data summarization to quantify the performance of a particular subset (Lin & Bilmes, 2011; Lindgren et al., 2016), and in particular SC has been used to model the data summarization problem (Mirzasoleiman et al., 2015). In many of these applications, the items of the dataset can be classified into a number of categories over which a balanced representation in the summary would be desirable. For example, images may be people of different nationalities, or news article documents may correspond to different perspectives on an issue. The FSC formulation emphasizes maintaining a certain information threshold via the constraint $f(S) \geq \tau$, balancing categories through a fairness constraint, while minimizing the summary size. One particular possibility

for the constraint on $f$ is that it be nearly its maximum value, i.e., we desire a summary that is as small as possible while maintaining nearly all of the information from the original dataset. We also note that data summarization motivates FSM (Halabi et al., 2022). In fact, for many use cases such a formulation of FSC may be more meaningful than the alternative FSM formulation, which places emphasis on restricting the summary to a particular budget.

**Training data subset selection.** A second application is training data subset selection in machine learning, which has previously been studied from the submodular optimization perspective (Wei et al., 2015). The fairness constraint can be used to address the problem of innacuracies introduced by class imbalance during model training, which is a major concern (Shwartz-Ziv et al., 2023). This problem can be modeled by FSC, where the constraint on the $f$ value represents a requirement that the smaller training data represent the overall dataset sufficiently well therefore promoting an accurate learned model, that the training data be balanced over the classes, and we seek to select the smallest amount of training data possible as training is computationally expensive.

**Neural network pruning and quantization.** Due to the massive size of modern neural networks, large, highly-accurate models may require multiple GPUs to do the inference, which limits the usability of such models. Neural network pruning and quantization address this by reducing network size or memory while preserving performance. In this task, the goal is to select a subset of neurons while preserving network performance. One common approach used in neural network pruning and quantization is the reweighted input change pruning (Frantar et al., 2022), where the objective function has been proved to preserve weak submodularity (Halabi et al., 2022). Assume that our goal is to prune the network with the objective function to achieve $70\%$ of the value of $F$ on the ground set i.e., $F(S) \geq 0.7\tau$, where $\tau$ is $F(U)$ where $U$ is the ground set. Additionally, a fairness constraint can ensure that neurons are proportionally selected across different layers or blocks to maintain structural balance.

**Influence maximization in social network analysis.** As a final application, influence maximization is an important problem in social network analysis. Suppose the social graph is described by $G = (V, E, \bar{\mathbf{w}})$, where $V$ is the set of nodes with $|V| = n$, $E$ denotes the set of edges, and $\bar{\mathbf{w}}$ is the weight vector defined on the set of edges $E$. The objective function is defined on subsets of the nodes of the graph to be the expected number of nodes influenced in the graph by a chosen seed set S, and it is well-known to be an example of a monotone and submodular function (Kempe et al., 2015). In this application, FSC addresses the problem where we want to find the subset of minimum size that could influence a target fraction of nodes, $f(S) \geq \tau$, while ensuring fairness in node selection based on associated features (e.g., demographics). This application highlights FSC's ability to balance influence spread across diverse subgroups in social networks.

## B.2 COMPARISON WITH EXISTING REDUCTIONS

In this section, we briefly discuss the difference between our reduction and the existing reduction from general submodular maximization to submodular cover.

The standard reduction from an instance of submodular cover (SC) with objective $f$ and threshold constraint $\tau$ to an instance of submodular maximization (SM) with objective $f$ and budget $k$ involves iteratively doing the following procedure: A guess is made for the size of the optimal solution ($|OPT|$) to the instance of SC, and this guess as the budget along with $f$ are input into an algorithm for SM. The procedure is repeated with increasingly large guesses until a solution is found with $f$ value sufficiently close to $\tau$. This process is relatively straightforward since the two problems are dual to each other, and the conversion requires only a flip between the objective and the constraint. A clear description of the process is provided in Iyer & Bilmes (2013a).

In contrast, in our fairness setting, the conversion from fair submodular cover (FSC) to fair submodular maximization (FSM) is less clear because of the more complex matroid structure of the fairness constraints. It is important to note that there is currently no existing formulation of the Submodular Cover (SC) problem that incorporates a matroid constraint, nor has any conversion process been developed to address such constraints. To address this, we devised a method where each guess of the size of the optimal solution for the instance of SC is used to construct an extended fairness matroid (we propose the concept of an extension to a matroid in Section 1.2). This matroid is then

input as a constraint into a bicriteria FSM algorithm, such as those developed in Sections 3.1 and 3.2. Furthermore, post-processing (Lines 6–12 in Algorithm 1 and Lines 7-13 in Algorithm 4) is required for each guess to ensure the fairness constraint is met, unlike the non-fairness setting where no such post-processing is necessary. A final difference between our fair setting and the general setting is that we are the first to introduce a converting process for multilinear extension algorithms (Algorithm 4).

### B.3 COMPARISON WITH ALGORITHMS FOR SM WITH MATROID CONSTRAINTS

In this section, we provide a comparison between our fair submodular maximization algorithms and existing algorithms for submodular maximization under matroid constraints.

First, we note that the fair submodular maximization (FSM) problem can be converted into an instance of submodular maximization with a general matroid constraint, as shown by Lemma 3 in Halabi et al. (2020). Therefore, existing algorithms for SM with a general matroid constraint can also be applied to FSM. Such algorithms return a feasible solution to the instance of SM with a matroid constraint, and achieve an approximation guarantee of at best $1 - 1/e$. However, our goal with our algorithms for FSM is to use them as subroutines in the conversion process, and so we develop algorithms that find sets that are better approximations to the optimal solution than $1 - 1/e$ but are not necessarily feasible. i.e., we propose algorithms for FSM with bicriteria approximation ratio. To this end, we have developed the notion of an extension of the fairness matroid (see Definition 2 in Section 1.2) and shown that by transferring to this "bigger" constraint we can achieve better objective values. In the case of our continuous algorithm, this means traveling within an extended polytope of the $\beta$-extension of the fairness matroid. This approach enables the $f$ value of the solution set to approach arbitrarily close to the optimal objective.

### B.4 OMITTED DISCUSSION ON THE FEASIBILITY OF FSC

In Section 1.2, we present the problem definition for FSC. Notice that for some input values of $\tau$, and $p_c, q_c$, there might be no feasible solution, i.e. the instance of FSC is invalid. Further, it might not be easy to check whether there exists a feasible solution with the value of submodular objective $f$ to be higher than $\tau$ and also satisfy the fairness constraint. However, if we have an algorithm that is guaranteed to produce an approximately feasible solution to FSC assuming the instance is valid, we can use this algorithm to check whether there exists an approximately feasible solution to the instance or not. In particular, suppose we have an algorithm for valid instances of FSC that is guaranteed to produce a fair solution $X$ such that $f(X) \geq \gamma\tau$ for some value $\gamma$ that is close to 1. Then if we run this algorithm on the instance, we can check if the returned solution satisfies $f(X) \geq \gamma\tau$ and is fair, in which case we know there exists a nearly feasible solution. If the solution does not satisfy the guarantees, then the FSC instance must not have a feasible solution at all. The algorithms proposed in this paper are examples of algorithms that provide nearly feasible solutions to valid instances of FSC. Another approach to find a value for $\tau$ which makes the FSC problem feasible is as follows. We can choose a value of the solution set, denoted as $r$, which satisfies that $q_c r \leq |U_c|$ for each $c \in [N]$, and run any non-bicriteria algorithm for the FSM instance with the fairness matroid $\mathcal{M}_{fair}(P, r, \vec{p}r, \vec{q}r)$. Since that $q_c r \leq |U_c|$, the rank of the fairness matroid is $r$. Therefore, if we set $\tau$ to be $f(S)$ where $S$ is the output solution set of the algorithm for FSM, then the FSC problem would be feasible since $S$ would be a feasible set for the FSC instance.

## C APPENDIX FOR SECTION 2

In this section, we present missing discussions and proofs from Section 2 in the main paper. We first present missing proofs of Theorem 1 about algorithm `convert-fair` in Section C.1. Then we present the proof of Theorem 2 about the converting algorithm `convert-continuous` for continuous algorithms in Section C.2. In addition, pseudocode for the algorithm `convert-continuous` is presented in Algorithm 4.

## C.1 PROOF OF THEOREM 1

In this section, we present the missing proofs of the lemmas that are used in the proof of Theorem 1. In order to prove Theorem 1, we need the following two lemmas. Lemma 4 guarantees that the solution set $S$ after the rounding step satisfies the fairness constraint for cover. Lemma 5 implies the inclusion relationship of the fairness matroid with the same fairness ratios.

**Lemma 4.** *For each guess $\kappa$ such that $\kappa \leq (1+\alpha)|OPT|$, the solution set $S$ in Algorithm 1 satisfies*

$$\beta\lfloor \frac{p_c|S|}{\beta} \rfloor \leq |S \cap U_c| \leq \beta\lceil \frac{q_c|S|}{\beta} \rceil, \qquad |S| = \beta\kappa.$$

*Proof.* Here we denote the solution set returned by the bicriteria algorithm for FSM as $S'$, and the solution set after the rounding steps from Line 6 to Line 8 as $S''$. From the definition of bicriteria approximation algorithm for FSM, we can see that the solution set returned by the subroutine algorithm for FSM satisfies that

$$|S' \cap U_c| \leq \beta\lceil q_c\kappa \rceil$$
$$\sum_{c \in [N]} \max\{|S' \cap U_c|, \beta\lfloor p_c\kappa \rfloor\} \leq \beta\kappa$$

After the rounding steps for each group from Line 6 to Line 8, the solution set satisfies that $|S'' \cap U_c| = \max\{\beta\lfloor p_c\kappa \rfloor, |S' \cap U_c|\}$ for any $c \in [N]$. It then follows that $\beta\lfloor p_c\kappa \rfloor \leq |S'' \cap U_c| \leq \beta\lceil q_c\kappa \rceil$. Since that $\sum_{c \in [N]} \max\{|S' \cap U_c|, \beta\lfloor p_c\kappa \rfloor\} \leq \beta\kappa$, we have that

$$|S''| = \sum_{c \in [N]} |S'' \cap U_c| = \sum_{c \in [N]} \max\{|S' \cap U_c|, \beta\lfloor p_c\kappa \rfloor\} \leq \beta\kappa.$$

From the assumption that $\sum_{c \in [N]} q_c \geq 1$ and $\sum_{c \in [N]} \min\{q_c, \frac{|U_c|}{\beta(1+\alpha)|OPT|}\} \geq 1$, after the second rounding phase from Line 10 to Line 12, we have $|S| = \beta\kappa$ and that for each group $c$,

$$\beta\lfloor p_c\kappa \rfloor \leq |S \cap U_c| \leq \beta\lceil q_c\kappa \rceil.$$

Since the solution set $S$ is of cardinality $\beta\kappa$, then we have

$$\beta\lfloor \frac{p_c|S|}{\beta} \rfloor \leq |S \cap U_c| \leq \beta\lceil \frac{q_c|S|}{\beta} \rceil.$$

$\square$

**Lemma 5.** *For any positive integers $\kappa_1, \kappa_2$ such that $\kappa_1 \leq \kappa_2$, we have that*

$$\mathcal{M}_{fair}(P, \kappa_1, \lceil \vec{p}\kappa_1 \rceil, \lceil \vec{q}\kappa_1 \rceil) \subseteq \mathcal{M}_{fair}(P, \kappa_2, \lfloor \vec{p}\kappa_2 \rfloor, \lceil \vec{q}\kappa_2 \rceil)$$

*Proof.* The lemma is equivalent to prove that for any subset $A \in \mathcal{M}_{fair}(P, \kappa_1, \lceil \vec{p}\kappa_1 \rceil, \lceil \vec{q}\kappa_1 \rceil)$, we have that $A$ is also in $\mathcal{M}_{fair}(P, \kappa_2, \lfloor \vec{p}\kappa_2 \rfloor, \lceil \vec{q}\kappa_2 \rceil)$. Since $\kappa_1 \leq \kappa_2$, $|A \cap U_c| \leq \lceil q_c\kappa_1 \rceil \leq \lceil q_c\kappa_2 \rceil$. For the second constraint, notice that $\sum_{c \in [N]} \max\{|A \cap U_c|, \lceil p_c\kappa_1 \rceil\} \leq \kappa_1$ is equivalent to that $\sum_{c \in [N]} \max\{|A \cap U_c|/\kappa_1, \frac{\lceil p_c\kappa_1 \rceil}{\kappa_1}\} \leq 1$. It then follows that

$$\sum_{c \in [N]} \max\{|A \cap U_c|/\kappa_2, \frac{\lfloor p_c\kappa_2 \rfloor}{\kappa_2}\} \leq \sum_{c \in [N]} \max\{|A \cap U_c|/\kappa_1, \frac{\lceil p_c\kappa_1 \rceil}{\kappa_1}\} \leq 1.$$

Therefore, $A \in \mathcal{M}_{fair}(P, \kappa_2, \lfloor \vec{p}\kappa_2 \rfloor, \lceil \vec{q}\kappa_2 \rceil)$. $\square$

We now prove Theorem 1.

**Theorem 1.** *Suppose $\sum_{c \in [N]} \min\{q_c, \frac{|U_c|}{\beta(1+\alpha)|OPT|}\} \geq 1$. Then any $(\gamma, \beta)$-bicriteria approximation algorithm for FSM that returns a solution set in time $\mathcal{T}(n, \kappa)$ can be converted into an approximation algorithm for FSC that is a $((1+\alpha)\beta, \gamma)$-bicriteria approximation algorithm that runs in time $O(\frac{\log(|OPT|)}{\log(\alpha+1)}\mathcal{T}(n, (1+\alpha)|OPT|))$.*

*Proof.* Denote the optimal solution of the FSC as $OPT$. Since by Lemma 4, the fairness constraint for cover is always satisfied. When the guess of $OPT$ satisfies that $|OPT| < \kappa \leq (1+\alpha)|OPT|$, by the definition of bicriteria approximation algorithm for FSM, it follows that

$$f(S) \geq \gamma \max_{X \in \mathcal{M}_{fair}(P, \kappa, \lfloor \vec{p}\kappa \rfloor, \lceil \vec{q}\kappa \rceil)} f(X).$$

Since $\kappa > |OPT|$, by Lemma 5, we have that $\mathcal{M}_{fair}(P, |OPT|, \lceil \vec{p}|OPT| \rceil, \lceil \vec{q}|OPT| \rceil) \subseteq \mathcal{M}_{fair}(P, \kappa, \lfloor \vec{p}\kappa \rfloor, \lceil \vec{q}\kappa \rceil)$. Therefore, it follows that

$$\max_{X \in \mathcal{M}_{fair}(P, \kappa, \lfloor \vec{p}\kappa \rfloor, \lceil \vec{q}\kappa \rceil)} f(X) \geq \max_{X \in \mathcal{M}_{fair}(P, |OPT|, \lceil \vec{p}|OPT| \rceil, \lceil \vec{q}|OPT| \rceil)} f(X)$$

Since $OPT$ is the optimal solution of FSC, we have that $\lceil p_c|OPT| \rceil \leq |OPT \cap U_c| \leq \lceil q_c|OPT| \rceil$. It implies that $OPT \in \mathcal{M}_{fair}(P, |OPT|, \lceil \vec{p}|OPT| \rceil, \lceil \vec{q}|OPT| \rceil)$. Therefore we can get

$$\max_{X \in \mathcal{M}_{fair}(P, |OPT|, \lceil \vec{p}|OPT| \rceil, \lceil \vec{q}|OPT| \rceil)} f(X) \geq f(OPT) \geq \tau.$$

Then

$$f(S) \geq \gamma\tau.$$

This means that the algorithm stops by the time when $\kappa$ reaches the region of $(|OPT|, (1+\alpha)|OPT|]$, which implies that the output solution set satisfies $|S| = \beta\kappa \leq \beta(1+\alpha)|OPT|$. Since by Lemma 4, the fairness constraint is always satisfied, the output solution set satisfies a $((1+\alpha)\beta, \gamma)$-approximation ratio. The algorithm makes $O(\log_{1+\alpha} |OPT|)$ calls to the bicriteria algorithm for FSM with $\kappa = 1+\alpha, (1+\alpha)^2, ..., (1+\alpha)|OPT|$, so the query complexity is $O(\sum_{i=1}^{\frac{\log(|OPT|)}{\log(\alpha+1)}} \mathcal{T}(n, (1+\alpha)^i))$. $\qquad\square$

## C.2 CONVERTING THEOREM FOR CONTINUOUS ALGORITHMS

In this section, we present and analyze the converting algorithm for the continuous algorithms, which is denoted as `convert-continuous`. The algorithm description is in Algorithm 4. The main result of the algorithm is presented in Theorem 2, which we restate as follows.

**Theorem 2.** *Any continuous algorithm with a $(\gamma, \beta)$-bicriteria approximation ratio for FSM that returns a solution in time $\mathcal{T}(n, \kappa)$ with probability at least $1 - \frac{\delta}{n}$ can be converted into an approximation algorithm for FSC such that with probability $1 - \delta$, the algorithm satisfies a $((1+\alpha)\beta, \frac{(1-3\varepsilon)\gamma - 2\varepsilon}{1+3\varepsilon+\frac{2\varepsilon}{\gamma}})$-bicriteria approximation ratio where $\frac{(1-3\varepsilon)\gamma-2\varepsilon}{1+3\varepsilon+\frac{2\varepsilon}{\gamma}}$ holds in expectation. The query complexity is at most $O(\log_{1+\alpha}|OPT|\mathcal{T}(n, (1+\alpha)|OPT|)) + \frac{n\log_{1+\alpha}|OPT|}{\varepsilon^2}\log\frac{n}{\delta})$.*

*Proof.* Throughout the proof, we use $OPT$ to denote the optimal solution of the FSM. In addition, we denote the optimal solution of FSM under the total cardinality $\kappa$ as $OPT_\kappa$, i.e., $OPT_\kappa = \arg\max_{S \in \mathcal{M}_{fair}(P, \kappa, \vec{p}\kappa, \vec{q}\kappa)} f(S)$. First of all, notice that there are at most $\min\{n, \log_{1+\alpha}|OPT| + 1\}$ number of guesses of $|OPT|$ before $\kappa$ reaches $|OPT| \leq \kappa \leq (1+\alpha)|OPT|$. By taking a union bound over all guess of $|OPT|$ we would obtain with probability at least $1 - \frac{\delta}{2}$ and for each guess of $|OPT|$, the algorithm for FSM outputs a solution **x** with a bicriteria approximation ratio of $(\gamma, \beta)$.

Since $\mathbf{F}(\mathbf{x}) \leq n\max_{s \in \mathcal{M}_\beta} f(s) \leq nf(OPT_\kappa)$, by the Chernoff bound in Lemma 8 and taking the union bound, it follows that with probability at least $1 - \frac{\delta}{2}$, for each guess of $|OPT|$, the estimate of $\mathbf{F}(\mathbf{x})$ in Line 4 of Algorithm 4 denoted as $Y$, satisfies that

$$|Y - \mathbf{F}(\mathbf{x})| \leq 2\varepsilon f(OPT_\kappa) + 3\varepsilon \mathbf{F}(\mathbf{x}).$$

By the definition of the bicriteria approximation ratio, it follows that $Y \geq \{(1-3\varepsilon)\gamma - 2\varepsilon\}f(OPT_\kappa)$.

Similar to the proof of Theorem 1, we can see that when $\kappa$, which is the guess of the size $OPT$ satisfies that $|OPT| \leq \kappa \leq (1+\alpha)|OPT|$, it follows that

$$f(OPT_\kappa) \geq \tau.$$

---

**Algorithm 4** `convert-continuous`

---

**Input**: An FSC instance with threshold $\tau$, fairness parameters $\vec{p}, \vec{q}$, partition $P$, a $(\gamma, \beta)$-bicriteria approximation algorithm for FSM, $\alpha > 0$

**Output**: $S \subseteq U$

1: $\kappa \leftarrow \lceil 1 + \alpha \rceil$, $S \leftarrow \emptyset$.
2: **while** true **do**
3:      $\mathbf{x} \leftarrow (\gamma, \beta)$-bicriteria approximation for FSM with fairness matroid $\mathcal{M}_{fair}(P, \kappa, \vec{p}\kappa, \vec{q}\kappa)$
4:      $Y \leftarrow$ average over $\frac{n}{2\varepsilon^2} \log(\frac{4n}{\delta})$ samples from $\mathbf{F}(\mathbf{x})$
5:      **if** $Y \geq \{(1 - 3\varepsilon)\gamma - 2\varepsilon\}\tau$ **then**
6:          $S \leftarrow$ pipage rounding of $\mathbf{x}$
7:          **for** $c \in [N]$ **do**
8:              **if** $|S \cap U_c| < p_c \beta \kappa$ **then**
9:                  Add new elements from $U_c/S$ to $S$ until $|S \cap U_c| \geq p_c \beta \kappa$
10:          **if** $|S| < \beta \kappa$ **then**
11:              **for** $c \in [N]$ **do**
12:                  **while** $|S| < \beta \kappa$ and $|S \cap U_c| < q_c \beta \kappa$ **do**
13:                      Add new elements in $U_c/S$ to $S$
14:      $\kappa \leftarrow \lceil (1 + \alpha)\kappa \rceil$
15: **return** $S$

---

Therefore, $Y \geq \{(1 - 3\varepsilon)\gamma - 2\varepsilon\}\tau$. It then follows that the algorithm stops before the guess of $|OPT|$ satisfies $|OPT| \leq \kappa \leq (1 + \alpha)|OPT|$. The value of multi-linear extension of the output fractional solution then satisfies

$$(1 + 3\varepsilon)\mathbf{F}(\mathbf{x}) + 2\varepsilon f(OPT_\kappa) \geq Y \geq \{(1 - 3\varepsilon)\gamma - 2\varepsilon\}\tau.$$

Combining the above inequality with that $\mathbf{F}(\mathbf{x}) \geq \gamma f(OPT_\kappa)$, then

$$\mathbf{F}(\mathbf{x}) \geq \frac{(1 - 3\varepsilon)\gamma - 2\varepsilon}{1 + 3\varepsilon + \frac{2\varepsilon}{\gamma}}\tau.$$

Since $\mathbf{x} \in \mathcal{P}(\mathcal{M}_\beta)$, where $\mathcal{M}_\beta$ is the $\beta$ extension of the fairness matroid under the guess $\kappa$, then after the pipage rounding step, we would have that $S \in \mathcal{M}_\beta$, and the value of objective function satisfies $\mathbb{E}f(S) \geq \mathbf{F}(\mathbf{x}) \geq \frac{(1-3\varepsilon)\gamma-2\varepsilon}{1+3\varepsilon+\frac{2\varepsilon}{\gamma}}\tau$. After the rounding steps from Line 7 to Line 13 in Algorithm 4, we would get that the final solution set satisfies

$$\mathbb{E}f(S) \geq \frac{(1 - 3\varepsilon)\gamma - 2\varepsilon}{1 + 3\varepsilon + \frac{2\varepsilon}{\gamma}}\tau$$

$$\beta \lfloor \frac{p_c|S|}{\beta} \rfloor \leq |S \cap U_c| \leq \beta \lceil \frac{q_c|S|}{\beta} \rceil$$

$$|S| \leq (1 + \alpha)\beta|OPT|.$$

$\square$

# D    APPENDIX FOR SECTION 3

In this section, we present the missing content in Section 3 in the main paper.

## D.1    APPENDIX FOR SECTION 3.1

In this portion of appendix, we present the missing details and proofs in Section 3.1 in the main paper, which is about two discrete algorithms `greedy-fair-bi` and `threshold-fairness-bi`. We begin by presenting the proof of Lemma 2, followed by proofs of the threshold greedy algorithm `threshold-fairness-bi`. Finally, the pseudocode

of `greedy-fair-bi` and `threshold-fairness-bi` are presented in Algorithm 5 and Algorithm 6 respectively.

First of all, we prove Lemma 2, which builds the relationship between the original fairness matroid and its $\beta$-extension for any $\beta \in \mathbb{N}_+$.

**Lemma 2.** *For any $\beta \in \mathbb{N}_+$ and any fairness matroid $\mathcal{M}_{fair}(P, \kappa, \vec{l}, \vec{u})$, denote $\mathcal{M}_\beta$ as the $\beta$-extended fairness matroid of $\mathcal{M}_{fair}(P, \kappa, \vec{l}, \vec{u})$. Then for any set $S \in \mathcal{M}_\beta$ with $|S| = \beta\kappa$, $T \in \mathcal{M}_{fair}(P, \kappa, \vec{l}, \vec{u})$ with $|T| = \kappa$, and any permutation of $S = (s_1, s_2, ..., s_{\beta\kappa})$, there exist a sequence $E = (e_1, e_2, ..., e_{\beta\kappa})$ such that each element in $T$ appears $\beta$ times in $E$ and that*

$$S_i \cup \{e_{i+1}\} \in \mathcal{M}_\beta, \qquad \forall i \in \{0, 1, ..., \beta\kappa\}$$

*where $S_i = (s_1, s_2, ..., s_i)$ and $S_0 = \emptyset$.*

*Proof.* Before proving the lemma, we define some notations here. For any sequence of any length $m$ denoted as $A = (a_1, a_2, ..., a_m)$, we define the number of element $x$ in the sequence as $|A^x|$, i.e., $|A^x| := |\{i : a_i = x\}|$. In addition, we define the number of elements of group $c$ in the sequence as $|A^c|$, i.e., $|A^c| = |\{i : a_i \in U_c\}| = \sum_{x \in U_c} |A^x|$. For the sequence $E$, we denote the sequence containing $i$-th element to the last elements as $E_i$, i.e., $E_i = (e_i, e_{i+1}, ..., e_{\beta\kappa})$. Now we prove a stronger claim which would imply the results in the lemma.

**Claim 1.** *For any $\beta \in \mathbb{N}_+$, denote $\mathcal{M}_\beta$ as the $\beta$-extension of the fairness matroid. Then for any set $S \in \mathcal{M}_\beta$, there exists a sequence $E = (e_1, ..., e_{\beta\kappa})$ such that for each $i \in \{0, 1, ..., \beta\kappa\}$, the sequence $F_i = (S_i, E_{i+1}) = (s_1, s_2, ..., s_i, e_{i+1}, ..., e_{\beta\kappa})$ satisfies that*

$$|F_i^c| \le u_c\beta \qquad \forall c \in [N]$$

$$\sum_{c \in [N]} \max\{|F_i^c|, l_c\beta\} \le \beta\kappa.$$

*Here for each $e \in E$, we have $e \in T$. Besides, we have that for any element $x \in T$,*

$$|F_i^x| \le \beta.$$

We prove the claim by induction. First, when $i = \beta\kappa$, $F_i = S$. Since $S \in \mathcal{M}_\beta$, the claim holds. Suppose the result in the claim holds for $i$, and we prove the claim for $i - 1$. There are two cases.

- Case 1. There exists some group $c_0$ such that $|(S_{i-1}, E_{i+1})^{c_0}| \le l_{c_0}\beta - 1$. Since $|T| = \kappa$, $|T \cap U_c| \ge l_c$ for each $c \in [N]$. Therefore, in this case, there exists at least one element $x \in U_{c_0} \cap T$ such that $|(S_{i-1}, E_{i+1})^x| < \beta$. Then choose $e_i = x$ and $E_i = (x, E_{i+1})$, the results in the claim will be satisfied.

- Case 2. For all group $c \in [N]$, $|(S_{i-1}, E_{i+1})^c| \ge l_c\beta$. Since the sequence $(S_{i-1}, E_{i+1})$ is of length $\beta\kappa - 1$, we have that

$$|(S_{i-1}, E_{i+1})| < \beta\kappa \le |T|\beta.$$

Therefore, there exists at least one group $c_1$ such that $|(S_{i-1}, E_{i+1})^{c_1}| < |T \cap U_{c_1}|\beta$. (Otherwise $\sum_{c \in [N]} |(S_{i-1}, E_{i+1})^c| \ge \sum_{c \in [N]} |T \cap U_c|\beta = \beta\kappa$, which breaks the assumption.) From $|(S_{i-1}, E_{i+1})^{c_1}| < |T \cap U_{c_1}|\beta$, we have that there exists at least one element $x \in T \cap U_{c_1}$ such that

$$|(S_{i-1}, E_{i+1})^x| \le \beta - 1.$$

Then we set the $i$-th element in $E$ to be $x$, then $E_i = (x, E_{i+1})$. It follows that $|(S_{i-1}, E_i)^x| \le \beta$. For each element $x' \in T/\{x\}$, $|(S_{i-1}, E_i)^{x'}| = |(S_{i-1}, E_{i+1})^{x'}| \le \beta$. Since $e_i = x \in U_{c_1}$. For group $c \ne c_1$, $|(S_{i-1}, E_i)^c| = |(S_{i-1}, E_{i+1})^c| \le u_c\beta$ by the assumption that the claim holds for iteration $i$. For group $c_1$, $|(S_{i-1}, E_i)^{c_1}| = |(S_{i-1}, E_i)^c| + 1 \le u_c\beta$. Since for all group $c \in [N]$, $|(S_{i-1}, E_{i+1})^c| \ge l_c\beta$, it follows that $|(S_{i-1}, E_i)^c| \ge l_c\beta$. Thus

$$\sum_{c \in [N]} \max\{|(S_{i-1}, E_i)^c|, l_c\beta\} = \sum_{c \in [N]} |(S_{i-1}, E_i)^c|$$

$$= |(S_{i-1}, E_i)| = \beta\kappa.$$

Thus we prove the claim for iteration $i-1$ under the assumption that the claim holds for $i$. By induction, the claim holds for all $i$. For $i = 0$, $(S_0, E_0) = E$. From the construction of $E$ we have that $|E| = \beta\kappa$, and that $|E^o| = \beta$ for all $o \in T$. Since for each group $c$, we have $|S_i \cup \{e_{i+1}\} \cap U_c| \leq |(S_i, E_i)^c|$. From the result in the claim, we can prove that $S_i \cup \{e_{i+1}\} \in \mathcal{M}_\beta$.

$\square$

### D.1.1 PROOF OF THEOREM 3

**Theorem 3.** *Suppose that* `greedy-fairness-bi` *is run for an instance of FSM, then* `greedy-fairness-bi` *outputs a solution $S$ that satisfies a $(1-\varepsilon, \frac{1}{\varepsilon})$-bicriteria approximation guarantee in at most $O\left(n\kappa/\varepsilon\right)$ queries of $f$.*

*Proof.* Denote the optimal solution of $\max_{S \in \mathcal{M}_{fair}(P,\kappa,\vec{l},\vec{u})} f(S)$ as $OPT$, i.e., $OPT = \arg\max_{S \in \mathcal{M}_{fair}(P,\kappa,\vec{l},\vec{u})} f(S)$. Since by Lemma 1, we have $\mathcal{M}_{1/\varepsilon} = \mathcal{M}_{fair}(P, \kappa/\varepsilon, \vec{l}/\varepsilon, \vec{u}/\varepsilon)$ is a matroid of rank $\kappa/\varepsilon$, then the Algorithm 5 ends after $\kappa/\varepsilon$ steps and the output solution set satisfies $|S| = \kappa/\varepsilon$. Since $S \in \mathcal{M}_{1/\varepsilon}$, then

$$|S \cap U_c| \leq u_c/\varepsilon \qquad \forall c \in [N]$$
$$\max_{c \in [N]}\{|S \cap U_c|, l_c/\varepsilon\} \leq \kappa/\varepsilon.$$

Then it remains to prove that $f(S) \geq (1-\varepsilon)f(OPT)$. From Lemma 2, we know that there exists a sequence $E$ that contains $1/\varepsilon$ copies of $OPT$ and that at each step $i$, $S_i \cup \{e_{i+1}\} \in \mathcal{M}_{1/\varepsilon}$. Then by the greedy selection strategy, we have

$$f(S_{i+1}) - f(S_i) \geq f(S_i \cup \{e_{i+1}\}) - f(S_i).$$

Thus by submodularity, we have

$$f(S_{i+1}) - f(S_i) \geq f(S_i \cup \{e_{i+1}\}) - f(S_i) \geq \Delta f(S, e_{i+1}).$$

Summing over all $i$, we would get

$$\sum_{i=0}^{\frac{k}{\varepsilon}-1} f(S_{i+1}) - f(S_i) \geq \sum_{i=0}^{\frac{k}{\varepsilon}-1} \Delta f(S, e_{i+1}).$$

Since the sequence $E$ contains $1/\varepsilon$ copies of each element in $OPT$, then $\sum_{i=0}^{\frac{k}{\varepsilon}-1} \Delta f(S, e_{i+1}) = 1/\varepsilon \sum_{o \in OPT} \Delta f(S, o)$. Since $\sum_{i=0}^{\frac{k}{\varepsilon}-1} f(S_{i+1}) - f(S_i) = f(S) - f(\emptyset)$ and that $f$ is nonnegative,

$$f(S) \geq \sum_{i=0}^{\frac{k}{\varepsilon}-1} \Delta f(S, e_{i+1}) \geq 1/\varepsilon \sum_{o \in OPT} \Delta f(S, o) \geq \frac{f(OPT) - f(S)}{\varepsilon}.$$

Thus we have

$$f(S) \geq \frac{1}{1+\varepsilon} f(OPT) \geq (1-\varepsilon)f(OPT).$$

$\square$

### D.1.2 PROOF OF THEOREM 4

Before we present the proof of the theorem, first we present the proof of the following lemma. Let us denote the solution set after the $i$-th element in `threshold-fairness-bi` as $S_i$. By Lemma 2, we know that we can construct a sequence $E = (e_1, e_2, .., e_{\kappa/\varepsilon})$ that contains $1/\varepsilon$ copies of $OPT$ and that $S_i \cup \{e_{i+1}\} \in \mathcal{M}_{1/\varepsilon}$. Then we have the following lemma.

**Lemma 6.** *For any $0 \leq i < \kappa/\varepsilon$, it follows that*

$$\Delta f(S_i, s_{i+1}) \geq (1 - \varepsilon)\Delta f(S_i, e_{i+1}) - \varepsilon d/\kappa.$$

*Proof.* First, we consider the case if $s_{i+1}$ is added to the solution set and is not a dummy variable, it follows that $\Delta f(S_i, s_{i+1}) \geq \tau$. Since $S_i \cup \{e_{i+1}\} \in \mathcal{M}_{1/\varepsilon}$, then if $e_{i+1} \notin S_i$, by submodularity we have $\Delta f(S_i, e_{i+1}) \leq \tau/(1 - \varepsilon)$. If $e_{i+1} \in S_i$, then $\Delta f(S_i, e_{i+1}) = 0 \leq \tau/(1 - \varepsilon)$. Next, we consider the case if $s_{i+1}$ is a dummy variable, then $\Delta f(S_i, s_{i+1}) = 0$. If $e_{i+1} \in S_i$, then $\Delta f(S_i, e_{i+1}) = 0$ and the above inequality in the lemma holds. If $e_{i+1} \notin S_i$, since $S_{\kappa_1} \cup \{e_{i+1}\} \in \mathcal{M}_{1/\varepsilon}$, then $\Delta f(S_i, e_{i+1}) \leq \varepsilon d/\kappa$. Therefore, we have that

$$\Delta f(S_i, s_{i+1}) \geq (1 - \varepsilon)\Delta f(S_i, e_{i+1}) - \varepsilon d/\kappa.$$

$\square$

Next, we present the proof of Theorem 4.

**Theorem 4.** *Suppose that* `threshold-fairness-bi` *is run for an instance of FSM with $\varepsilon \in (0, 1)$. Then* `threshold-fairness-bi` *outputs a solution $S$ that satisfies a $(1 - 2\varepsilon, \frac{1}{\varepsilon})$-bicriteria approximation guarantee in at most $O\left(n/\varepsilon \log(\kappa/\varepsilon)\right)$ queries of $f$.*

*Proof.* First, notice that the Algorithm 6 ends in at most $(1/\varepsilon)\log(\kappa/\varepsilon)$ number of iterations. Therefore, there are at most $n/\varepsilon \log(\kappa/\varepsilon)$ number of queries to $f$. Next, we prove the bicriteria approximation ratio of `threshold-fairness-bi`. From the description of Algorithm 6, we have that the output solution set $S \in \mathcal{M}_{1/\varepsilon}$, then

$$|S \cap U_c| \leq u_c/\varepsilon \qquad \forall c \in [N]$$
$$\max_{c \in [N]}\{|S \cap U_c|, l_c/\varepsilon\} \leq \kappa/\varepsilon.$$

It remains to prove that $f(S) \geq (1 - 2\varepsilon)f(OPT)$ where $OPT$ is defined as the optimal solution of FSM, i.e., $OPT = \arg\max_{S \in \mathcal{M}_{fair}(P, \kappa, \vec{l}, \vec{u})} f(S)$. For simplicity, we assume the returned solution has size $|S| = k_1$. As discussed in the proof of Theorem 3, $\text{Rank}(\mathcal{M}_{1/\varepsilon}) = \kappa/\varepsilon$. Here we denote the solution set as $S = (s_1, s_2, ..., s_{\kappa/\varepsilon})$, and we define $S_i$ as $S_i = (s_1, ..., s_i)$. Here $s_i$ is the $i$-th element added to the solution set. In the case when the threshold $\tau$ drops below $\varepsilon d/\kappa$ at the termination and $\kappa_1 \leq \kappa/\varepsilon$, we can add dummy elements to $S$ such that $|S| = \kappa/\varepsilon$. By Lemma 6, we have that there is a sequence $E = (e_1, e_2, ..., e_{k/\varepsilon})$ that contains $1/\varepsilon$ copies of $OPT$ and

$$\Delta f(S_i, s_{i+1}) \geq (1 - \varepsilon)\Delta f(S_i, e_{i+1}) - \varepsilon d/\kappa.$$

Thus by submodularity, we have

$$f(S_{i+1}) - f(S_i) \geq (1 - \varepsilon)\{f(S_i \cup \{e_{i+1}\}) - f(S_i)\} - \varepsilon d/\kappa \geq (1 - \varepsilon)\Delta f(S, e_{i+1}) - \varepsilon d/\kappa.$$

Summing over all $i$, we would get

$$\sum_{i=0}^{k/\varepsilon-1} f(S_{i+1}) - f(S_i) \geq (1 - \varepsilon)\sum_{i=0}^{\kappa/\varepsilon-1} \Delta f(S, e_{i+1}) - d.$$

Since the sequence $E$ contains $1/\varepsilon$ copies of each element in $OPT$, then $\sum_{i=0}^{\frac{k}{\varepsilon}-1} \Delta f(S, e_{i+1}) = 1/\varepsilon \sum_{o \in OPT} \Delta f(S, o)$. Since $\sum_{i=0}^{\kappa/\varepsilon-1} f(S_{i+1}) - f(S_i) = f(S) - f(\emptyset)$ and that $f$ is nonnegative,

$$f(S) \geq (1 - \varepsilon)\sum_{i=0}^{\kappa/\varepsilon-1} \Delta f(S, e_{i+1}) - d$$

$$\geq \frac{(1 - \varepsilon)}{\varepsilon}\sum_{o \in OPT} \Delta f(S, o) - d$$

$$\geq \frac{1}{\varepsilon}\{(1 - \varepsilon)\{f(OPT) - f(S)\} - \varepsilon f(OPT)\}$$

---

**Algorithm 5** `greedy-fairness-bi`

---

1: **Input:** $\varepsilon$, fairness matroid $\mathcal{M}_{fair}(P, \kappa, \vec{l}, \vec{u})$
2: **Output:** $S \in U$
3: $S \leftarrow \emptyset$
4: Denote $\mathcal{M}_{fair}(P, \kappa/\varepsilon, \vec{l}/\varepsilon, \vec{u}/\varepsilon)$ as $\mathcal{M}_{1/\varepsilon}$.
5: **while** $\exists x$ s.t. $S \cup \{x\} \in \mathcal{M}_{1/\varepsilon}$ **do**
6:      $V \leftarrow \{x \in U | S \cup \{x\} \in \mathcal{M}_{1/\varepsilon}\}$
7:      $u \leftarrow \arg\max_{x \in V} \Delta f(S, x)$
8:      $S \leftarrow S \cup \{u\}$
     **return** $S$

---

**Algorithm 6** `threshold-fairness-bi`

---

1: **Input:** $\varepsilon$, fairness matroid $\mathcal{M}_{fair}(P, \kappa, \vec{l}, \vec{u})$
2: **Output:** $S \in U$
3: $S \leftarrow \emptyset$
4: Denote $\mathcal{M}_{fair}(P, \kappa/\varepsilon, \vec{l}/\varepsilon, \vec{u}/\varepsilon)$ as $\mathcal{M}_{1/\varepsilon}$
5: $d \leftarrow \max_{\{x\} \in M_{1/\varepsilon}} f(\{x\})$
6: **for** $\tau = d; \tau \geq \varepsilon d/k; \tau \leftarrow \tau(1 - \varepsilon)$ **do**
7:      **for** $x \in U$ **do**
8:          **if** $S \cup \{x\} \in \mathcal{M}_{1/\varepsilon}$ and $\Delta f(S, x) \geq \tau$ **then**
9:              $S \leftarrow S \cup \{x\}$
10:          **if** $|S| = \kappa/\varepsilon$ **then**
11:              **return** $S$
     **return** $S$

---

$$\geq \frac{1}{\varepsilon}\{(1 - 2\varepsilon)f(OPT) - (1 - \varepsilon)f(S)\}.$$

By re-arranging the above equation, we have that

$$f(S) \geq (1 - 2\varepsilon)f(OPT).$$

$\square$

D.2    APPENDIX FOR SECTION 3.2

In this section, we provide the omitted content from Section 3.2 of the main paper. Specifically, we present the proof of Lemma 7, which offers the theoretical guarantee for the subroutine algorithm `decreasing-threshold-procedure` of the continuous algorithm `cont-bi`. The statement of the Lemma is as follows.

**Lemma 7.** *During each call of* `decreasing-threshold-procedure`*, the output coordinate set $B$ satisfies that*

$$\boldsymbol{F}(\boldsymbol{x} + \varepsilon \boldsymbol{1}_B) - \boldsymbol{F}(\boldsymbol{x}) \geq \varepsilon\{\ln(1/\varepsilon) + 1\}((1 - 6\varepsilon)f(OPT) - \boldsymbol{F}(\boldsymbol{x} + \varepsilon \boldsymbol{1}_B)).$$

*Proof.* For notation simplicity, we denote the rank of the matroid $\mathcal{M}_{\ln(\frac{1}{\varepsilon})+1}$ as $m$, i.e., $m := (\ln(\frac{1}{\varepsilon}) + 1)\kappa$. Here we denote the output solution set as $B = \{b_1, b_2, ..., b_m\}$ where $b_i$ is the $i$-th element that is added to set $B$. Here if $|B| < m$, then for any $i > |B|$, $b_i$ is defined as a dummy variable. In addition, we define $B_i = \{b_1, b_2, ..., b_i\}$. Since $\mathcal{M}_{\ln(\frac{1}{\varepsilon})+1}$ is an $\ln(\frac{1}{\varepsilon}) + 1$-extension of the original fairness matroid, then by Lemma 2, there exists a sequence $E = (e_1, e_2, ..., e_m)$ such that $E$ contains $\ln(1/\varepsilon) + 1$ copies of the optimal solution $OPT = \{o_1, o_2, ..., o_\kappa\}$ such that $B_{i-1} \cup \{e_i\} \in \mathcal{M}_{\ln(1/\varepsilon)+1}$ for each $i \in [m]$.

Notice that by Lemma 8, we have that with probability at least $1 - \frac{\varepsilon^3}{2n^4}$, for any fixed $\boldsymbol{x} + \boldsymbol{1}_B$ and fixed element $u$, the empirical mean $\hat{X}(B_i, u)$, which is the average over $\frac{3\kappa}{\varepsilon^2} \log \frac{4n^4}{\varepsilon^3}$ samples of the

random variable $X = \Delta f(S(\mathbf{x} + \varepsilon \mathbf{1}_{B_i}), u)$ satisfies that

$$|\hat{X}(B_i, u) - \mathbb{E}\Delta f(S(\mathbf{x} + \varepsilon \mathbf{1}_{B_i}), u)| \le \frac{\varepsilon}{\kappa} f(OPT) + \varepsilon \mathbb{E}\Delta f(S(\mathbf{x} + \varepsilon \mathbf{1}_{B_i}), u). \qquad (1)$$

Since during the execution of `cont-bi`, there are at most $\frac{n}{\varepsilon^2} \log(\kappa/\varepsilon)$ such estimations, by applying the union bound, we have that with probability at least $1 - \frac{1}{2n^2}$, the inequality (1) holds for each $\mathbf{x}$, $B$ and $u$ during `cont-bi`. From the description in Algorithm 3, we can see that $\hat{X}(B_{i-1}, b_i) \ge w$. For the element $e_i$, we have that $\hat{X}(B_{i-1}, e_i) \le \frac{w}{1-\varepsilon}$ or at the last iteration, we have that $\hat{X}(B_{i-1}, e_i) \le \frac{\varepsilon d}{\kappa}$. Therefore, we have that $\hat{X}(B_{i-1}, b_i) \ge (1 - \varepsilon)\hat{X}(B_{i-1}, e_i) - \frac{\varepsilon d}{\kappa}$. Since $OPT = \max_{S \in \mathcal{M}_{fair}(P, \kappa, \vec{p}\kappa, \vec{q}\kappa)} f(S) \ge d$, it then follows that

$$(1 + \varepsilon)\mathbb{E}\Delta f(S(\mathbf{x} + \varepsilon \mathbf{1}_{B_{i-1}}), b_i) \ge (1 - \varepsilon)^2 \mathbb{E}\Delta f(S(\mathbf{x} + \varepsilon \mathbf{1}_{B_{i-1}}), e_i) - \frac{3\varepsilon}{\kappa} f(OPT).$$

By rearranging the above inequality and simple calculations, we have

$$\mathbb{E}\Delta f(S(\mathbf{x} + \varepsilon \mathbf{1}_{B_{i-1}}), b_i) \ge (1 - 3\varepsilon)\mathbb{E}\Delta f(S(\mathbf{x} + \varepsilon \mathbf{1}_{B_{i-1}}), e_i) - \frac{3\varepsilon}{\kappa} f(OPT).$$

By the construction of set $B$, we would get

$$\begin{aligned}
\mathbf{F}(\mathbf{x} + \varepsilon \mathbf{1}_B) - \mathbf{F}(\mathbf{x}) &= \sum_{i=1}^{m} \mathbf{F}(\mathbf{x} + \varepsilon \mathbf{1}_{B_i}) - \mathbf{F}(\mathbf{x} + \varepsilon \mathbf{1}_{B_{i-1}}) \\
&= \sum_{i=1}^{m} \varepsilon \cdot \frac{\partial \mathbf{F}}{\partial b_i}\Big|_{x = \mathbf{x} + \mathbf{1}_{B_{i-1}}} \\
&\ge \varepsilon \sum_{i=1}^{m} \mathbb{E}\Delta f(S(\mathbf{x} + \varepsilon \mathbf{1}_{B_{i-1}}), b_i) \\
&\ge \varepsilon \sum_{i=1}^{m} (1 - 3\varepsilon)\mathbb{E}\Delta f(S(\mathbf{x} + \varepsilon \mathbf{1}_{B_{i-1}}), e_i) - \frac{3\varepsilon f(OPT)}{\kappa} \\
&= \varepsilon(1 - 3\varepsilon) \sum_{i=1}^{m} \mathbb{E}\Delta f(S(\mathbf{x} + \varepsilon \mathbf{1}_{B_{i-1}}), e_i) \\
&\quad - 3\varepsilon^2 (\ln(\frac{1}{\varepsilon}) + 1) f(OPT) \\
&\ge \varepsilon(1 - 3\varepsilon) \sum_{i=1}^{m} \mathbb{E}\Delta f(S(\mathbf{x} + \varepsilon \mathbf{1}_B), e_i) \\
&\quad - 3\varepsilon^2 (\ln(\frac{1}{\varepsilon}) + 1) f(OPT),
\end{aligned}$$

where the last inequality results from the submodularity of $f$. From Lemma 2, we have that

$$\begin{aligned}
\mathbf{F}(\mathbf{x} + \varepsilon \mathbf{1}_B) - \mathbf{F}(\mathbf{x}) &\ge \varepsilon(1 - 3\varepsilon)(\ln(\frac{1}{\varepsilon}) + 1) \sum_{i=1}^{\kappa} \mathbb{E}\Delta f(S(\mathbf{x} + \varepsilon \mathbf{1}_B), o_i) \\
&\quad - 3\varepsilon^2 (\ln(\frac{1}{\varepsilon}) + 1) f(OPT) \\
&\ge \varepsilon(1 - 3\varepsilon)(\ln(\frac{1}{\varepsilon}) + 1)\{f(OPT) - \mathbf{F}(\mathbf{x} + \varepsilon \mathbf{1}_B)\} \\
&\quad - 3\varepsilon^2 (\ln(\frac{1}{\varepsilon}) + 1) f(OPT) \\
&\ge \varepsilon\{\ln(1/\varepsilon) + 1\}((1 - 6\varepsilon) f(OPT) - \mathbf{F}(\mathbf{x} + \varepsilon \mathbf{1}_B)).
\end{aligned}$$

$\square$

### D.2.1 PROOF OF THEOREM 5

**Theorem 5.** *Suppose that Algorithm 2 is run for an instance of FSM, then with probability at least $1 - \frac{1}{n^2}$, `cont-thresh-greedy-bi` outputs a solution $S$ that satisfies a $(1 - 7\varepsilon, \ln(\frac{1}{\varepsilon}) + 1)$-bicriteria approximation guarantee in at most $O\left(\frac{n\kappa}{\varepsilon^4}\ln^2(n/\varepsilon)\right)$ queries of $f$.*

*Proof.* First of all, from the description of the subroutine algorithm `DTP` in Algorithm 3, we can see that there are at most $\log(\kappa/\varepsilon)/\varepsilon$ number of iterations in the outer for loop. Therefore, the subroutine algorithm `DTP` takes at most $O(\frac{n\kappa \ln(n/\varepsilon)\ln(\kappa/\varepsilon)}{\varepsilon^3})$. Since there are at most $\frac{1}{\varepsilon}$ calls to `DTP`, we can prove the sample complexity.

Next, we prove the bicriteria approximation ratio. By Definition 5, it is equivalent to prove that $\mathbf{x} \in \mathcal{P}(\mathcal{M}_{\ln(1/\varepsilon)+1})$ and $\mathbf{F}(\mathbf{x}) \geq (1 - 7\varepsilon)f(OPT)$. Denote $B^{(t)}$ to be the output set of the $t$-th call to the subroutine algorithm `DTP`. Then it follows that the output solution set $\mathbf{x}$ of `cont-bi` can be denoted as $\mathbf{x} = \sum_{t=1}^{1/\varepsilon} \varepsilon\mathbf{1}_{B^{(t)}}$. Since $B^{(t)} \in \mathcal{M}_{\ln(1/\varepsilon)+1}$, we have that $\mathbf{1}_{B^{(t)}} \in \mathcal{P}(\mathcal{M}_{\ln(1/\varepsilon)+1})$. By the fact that $\mathcal{P}(\mathcal{M}_{\ln(1/\varepsilon)+1})$ is convex, we have that $\mathbf{x} \in \mathcal{P}(\mathcal{M}_{\ln(1/\varepsilon)+1})$. Denote the fractional solution $\mathbf{x}$ after $t$-th step as $\mathbf{x}_t$, then by Lemma 7, we have

$$\mathbf{F}(\mathbf{x}_{t+1}) - \mathbf{F}(\mathbf{x}_t) \geq \varepsilon\{\ln(1/\varepsilon) + 1\}((1 - 6\varepsilon)f(OPT) - \mathbf{F}(\mathbf{x}_{t+1})).$$

For notation simplicity, we define $L = \varepsilon\{\ln(1/\varepsilon) + 1\}$. It then follows that

$$\mathbf{F}(\mathbf{x}_{t+1}) \geq \frac{\mathbf{F}(\mathbf{x}_t) + L(1 - 6\varepsilon)f(OPT)}{1 + L}$$

Since there are $1/\varepsilon$ iterations in `cont-bi`, the output $\mathbf{x}$ satisfies that $\mathbf{x} = \mathbf{x}_{1/\varepsilon}$. By applying induction to the above inequality, we would get

$$\begin{aligned}
\mathbf{F}(\mathbf{x}_{1/\varepsilon}) &\geq (1 - (1 + L)^{-1/\varepsilon})\{(1 - 6\varepsilon)f(OPT)\} \\
&\geq (1 - e^{\frac{1}{\varepsilon}(\frac{L^2}{2} - L)})\{(1 - 6\varepsilon)f(OPT)\} \\
&\geq (1 - \varepsilon)(1 - 6\varepsilon)f(OPT) \\
&\geq (1 - 7\varepsilon)f(OPT).
\end{aligned}$$

$\square$

Observe that compared to `greedy-fair-bi` and `threshold-fairness-bi`, our proposed algorithm `cont-bi` demonstrates an enhanced approximation ratio for the cardinality of the output solution set, improving from $O(1/\varepsilon)$ to $O(\ln(1/\varepsilon))$. This improvement is achieved while maintaining the same order of function value violation, specifically $f(S) \geq (1 - O(\varepsilon))f(OPT)$. However, this enhancement requires an increased number of queries. This suggests the potential for attaining comparable approximation ratios for the submodular cover problem under specific types of matroid-type constraints.

**Corollary 5.1.** *Using the Algorithm 3 as the subroutine for the converting algorithm in Algorithm 1, we obtain an algorithm that achieves an approximation ratio of $((1 + \alpha)\ln(\frac{1}{\varepsilon}) + 1, 1 - O(\varepsilon))$ in at most $O(\frac{n(1+\alpha)|OPT|\log^2(\frac{n}{\varepsilon})\log|OPT|}{\varepsilon^4\alpha})$ with high probability.*

*Proof.* The result of the approximation ratio can be obtained by combining Theorem 2 and Theorem 5 together. Here we provide proof of the sample complexity. Notice that for each guesses of OPT of cardinality $\kappa_g$, the algorithm `cont-bi` that runs with the input $\mathcal{M}_{fair}(P, \kappa_g, \vec{p}\kappa_g, \vec{u}\kappa_g)$ uses at most $O(\frac{n\kappa_g \log^2(\frac{n}{\varepsilon})}{\varepsilon^4})$. From the result in Theorem 2 and Theorem 5, the total number of sample complexity would be $O(\frac{n(1+\alpha)|OPT|\log^2(\frac{n}{\varepsilon})\log|OPT|}{\varepsilon^4\alpha})$. $\square$

## E PROOF OF TECHNICAL LEMMAS

**Lemma 8** (Relative + Additive Chernoff Bound (Lemma 2.3 in Badanidiyuru & Vondrák (2014)))**.** *Let $X_1, ..., X_N$ be independent random variables such that for each $i$, $X_i \in [0, R]$ and $\mathbb{E}[X_i] = \mu$ for all $i$. Let $\widehat{X}_N = \frac{1}{N}\sum_{i=1}^N X_i$. Then*

$$P(|\widehat{X}_N - \mu| > \alpha\mu + \varepsilon) \leq 2\exp\{-\frac{N\alpha\varepsilon}{3R}\}.$$

# F  GREEDY-BI ALGORITHM

---

**Algorithm 7** `greedy-bi`

---

1: **Input:** $\varepsilon, \tau$
2: **Output:** $S \in U$
3: $S \leftarrow \emptyset$
4: **while** $f(S) \leq (1 - \varepsilon)\tau$ **do**
5:      $u \leftarrow \arg\max_{x \in U} \Delta f(S, x)$
6:      $S \leftarrow S \cup \{u\}$
    **return** $S$

---

# G  ADDITIONAL EXPERIMENTS

## G.1  IMPLEMENTATION

**Maximum Coverage.** In maximum coverage problems, the objective is to identify a set of fixed nodes that optimally maximize coverage within a network or graph. Given a graph $G = (V, E)$, where $V$ and $E$ respectively represent the set of vertices and nodes in the graph. Define a function $N : V \to 2^V$ as $N(v) = \{u, (u, v) \in E\}$, which represents the collection of neighbors of node $v$. Then the objective of this maximization problem can be defined as the monotone submodular function $f(S) = | \cup_{v \in S} N(v)|$. The dataset utilized in our maximum coverage experiments include Twitch_2000 and the Twitch_5000 of Twitch Gamers, which is a uniformly sampled subgraph of the Twitch Gamers dataset (Rozemberczki & Sarkar, 2021), comprising 5,000 vertices and 2,000 vertices (users) who speak English, German, French, Spanish, Russian, or Chinese respectively. We aim to develop a solution with a high $f$ value exceeding a given threshold $\tau$ while ensuring a fair balance between users who speak different languages.

**Set Covering** For the datasets annotated with tags, the objective of set covering is to extract a diverse subset that maximizes the objective function $f(S) = | \cup_{x \in S} t(x)|$, where the function $t$ maps an element $x$ in a set $N$ to its corresponding tags $t(x)$. The dataset employed in our set covering experiments is a subset of Corel5k Duygulu et al. (2002). For each item in the dataset, we randomly added a category from $\{0, 2, 3, 4, 5\}$ with a probability of $0.5$. Any item not assigned a random category was assigned category $1$. By ensuring a balanced distribution of solutions across categories, we aim to extract a representative set with a high $f$ value that surpasses a given threshold $\tau$.

**Experimental setup for max coverage.** We implement our proposed algorithm on two different instances, including the Twitch_5000 dataset and Twitch_2000. The algorithms implemented include `convert-fair` leveraging two subroutines provided in Appendix D: `greedy-fair-bi` (Algorithm 5, referred to as "GREEDY-Fair") and `threshold-fairness-bi` (Algorithm 6, referred to as "GREEDY-Fair"). On the relatively small dataset Twitch_2000, we also implement `convert-continuous` using `cont-bi` as the subroutine ( referred to as "CONTI-Fair"). Due to high query complexity of the continuous algorithm `cont-bi`, we evaluate `convert-continuous` heuristically by taking 5 samples per estimation in Line 8 of the `decreasing-threshold-procedure`. We compare our approach to the greedy baseline, `greedy-bi`, provided in Appendix F.

To ensure a fair comparison based on the quality of the solutions, we use different default values for the parameter $\varepsilon$ in each algorithm. This is because each algorithm has a varying approximation ratio. Specifically, we set $\varepsilon = 0.1, \alpha = 0.2, u_c = 1.1/C, l_c = 0.9/C$ for `greedy-bi` and `greedy-fair-bi` (where $C$ is the number of groups). For `threshold-fairness-bi`, we use $\varepsilon = 0.05$ while keeping the other parameters the same.

All the experiments are conducted on a single machine equipped with a 13th Gen Intel(R) Core(TM) i7-13700 CPU, 32GB of RAM, and Ubuntu 22.04.3 LTS. Each experiment with one set of parameters can be done in 120 seconds.

## G.2 EXPERIMENTS SETUP FOR SET COVERING.

We implement our proposed algorithm `convert-fair` leveraging two subroutines provided in Appendix D: `greedy-fair-bi` (Algorithm 5) and `threshold-fairness-bi` (Algorithm 6). We compare our approach to the greedy baseline, `greedy-bi`, provided in Appendix F.

To ensure a fair comparison based on the quality of the solutions, we use different default values for the parameter $\varepsilon$ in each algorithm. This is because each algorithm has a varying approximation ratio. Specifically, we set $\varepsilon = 0.1, \alpha = 0.2, u_c = 1.1/C, l_c = 0.9/C$ for `greedy-bi` and `greedy-fair-bi` (where $C$ is the number of groups). For `threshold-fairness-bi`, we use $\varepsilon = 0.05$ while keeping the other parameters the same.

All the experiments are conducted on a single machine equipped with a 13th Gen Intel(R) Core(TM) i7-13700 CPU, 32GB of RAM, and Ubuntu 22.04.3 LTS. Each experiment with one set of parameters can be done in 30 seconds.

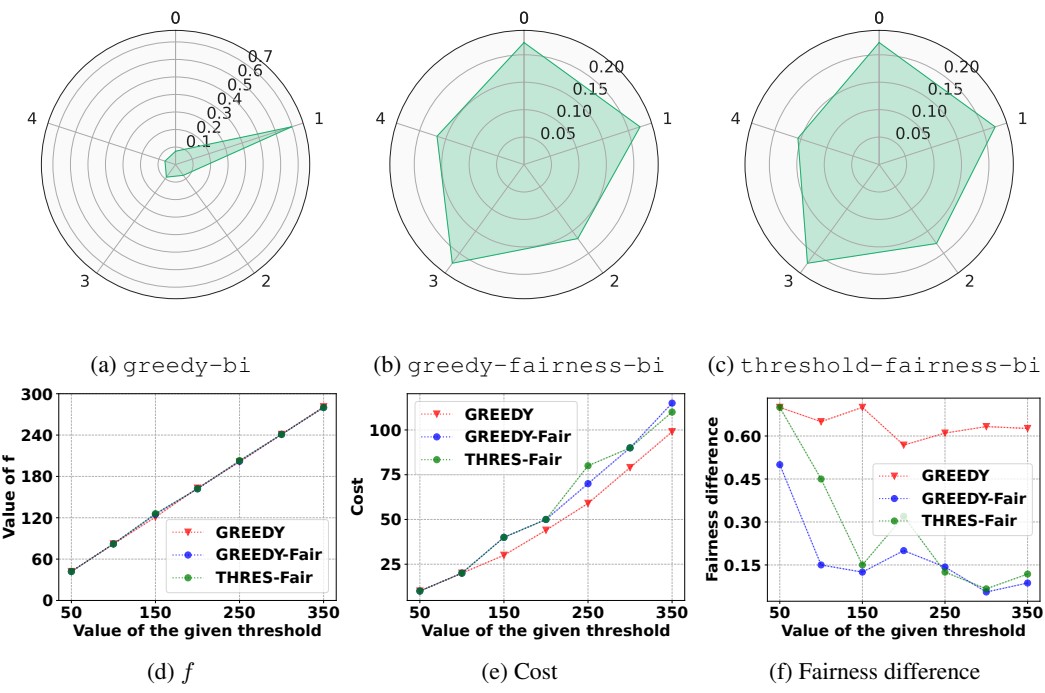

Figure 2: Performance comparison on the Corel dataset for Set Covering. 2a, 2b, 2c illustrate the distribution of images across various categories in the solutions produced by different algorithms with $\tau = 300$. $f$: the value of the objective submodular function. Cost: the size of the returned solution. Fairness difference: $(\max_c |S \cap U_c| - \min_c |S \cap U_c|)/|S|$
.

## G.3 RESULTS

## G.4 RESULTS ON COREL DATASET

Figures 2a, 2b and 2c showcase the distribution of images across various categories in the solutions produced by these algorithms with $\tau = 300$. Figures 2d, 2e and 2f present the performance of these algorithms ($f$ value, cost, and fairness difference) for varying values of $\tau$. As shown in Figure 2a, with $\tau = 300$, over 70% of the pictures in the solution returned by `greedy-bi` are labeled as category '1'. While the solutions produced by `greedy-fair-bi` and `threshold-fairness-bi` exhibit way fairer distributions across various categories as shown in Figures 2a and 2b). Similarly, as the value of given $\tau$ increases, the magnitude of this difference also increases (see 2f). Figure 2d showcases that for all these algorithms the objective function value $f(S)$ scales almost linearly with the threshold $\tau$, which aligns with the theoretical guarantees of the approximation ratio.

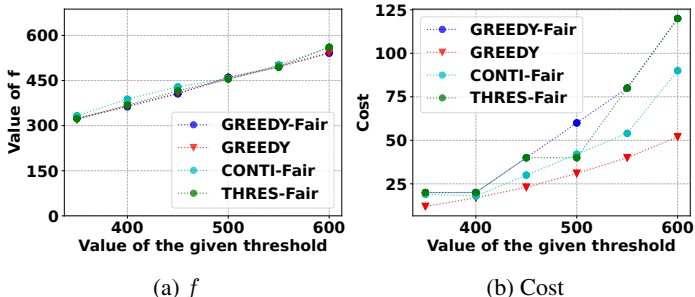

(a) $f$          (b) Cost

Figure 3: Performance comparison on the Twitch_2000 dataset for maximum coverage. "CONTI-Fair" corresponds to the `convert-fair` algorithm using `cont-bi` as the subroutine. $f$: the value of the objective submodular function. Cost: the size of the returned solution. Fairness difference: $(\max_c |S \cap U_c| - \min_c |S \cap U_c|)/|S|$

.

Notably, unlike the results presented in Section 4, our proposed algorithms achieve comparable costs to the `greedy-bi` solution (as shown in Figure 2e) on the Corel5k dataset. This is likely because the Corel5k dataset is less biased and the marginal gains for adding different elements are more uniform, compared to the Twitch Gamer dataset.

### G.5 RESULTS ON THE TWITCH_2000 DATASET

The results of comparing different algorithms on the Twitch_2000 dataset are presented in Figure 3a and 3b. From the results, we can see under fairness constraints, the continuous algorithm CONTI-Fair achieves a lower cost compared to the discrete algorithms, GREEDY-Fair and THRES-Fair, aligning with the theoretical guarantees presented in the main paper. However, CONTI-Fair incurs a higher cost than the greedy algorithm without fairness constraint, potentially due to the limited sampling (five samples per estimation) of the multilinear extension in Line 8 in Algorithm 3, which falls short of the theoretical requirements.

