# OpenReview forum: "Fair Submodular Cover"
_ICLR.cc/2025/Conference — ICLR 2025 Poster_

### Official Review · Reviewer_pFnE · 2024-11-01

**Soundness:** 2
**Presentation:** 2
**Contribution:** 2
**Rating:** 6
**Confidence:** 4

**Summary:**

This paper studies the submodular cover problem under the fairness constraint (FSC). The authors propose two algorithms, convert-fair and convert-continuous, which exploit the dual relationship between Fair Submodular Maximization (FSM) and FSC to transform bicriteria approximation algorithms for FSM into ones applicable to FSC. Next, they propose three algorithms for FSM that, when combined with their converting algorithms, produce approximate solutions for FSC. The discrete algorithms achieve approximation ratios of $(1 - O(\varepsilon),\frac{1}{\varepsilon})$ for FSM, while the continuous algorithm achieves a better ratio of $(1 - O(\varepsilon),\ln(\frac{1}{\varepsilon}) + 1)$ but needs more queries. By integrating these algorithms into their framework, they obtain improved FSC algorithms with better approximation ratios. The continuous algorithm for FSM with the converting algorithm achieves a $((1 + \alpha) \ln (n) + 1, 1 − 7/n)$ bicriteria approximation guarantee for an instance of FSC when  $\varepsilon = \frac{1}{n}$, which is close to the lower bound.

**Strengths:**

* The proposed algorithms are easy to follow.
* The proposed continuous algorithm approximately matches the theoretical lower bound, demonstrating its efficiency.

**Weaknesses:**

* The motivation for considering FSC is unclear. The authors only say that FSM has been well studied while FSC has not. This can not be a good motivation.
* The experiments demonstrate the advantages of the discrete algorithms on the FSC problem compared to the standard greedy algorithm; however, the solutions obtained by the discrete algorithms include many more elements. Moreover, the experiments lack the presentation of the effectiveness of the continuous algorithm.

**Questions:**

* What are the motivations for FSC, e.g., real-world applications?
* Compared to the conversion algorithms without fairness constraints, what is the novelty of the conversion algorithms proposed in the paper?
* In Theorem 2, the approximation ratio $\frac{(1 - 3\varepsilon)\gamma - 2\varepsilon}{1 + 3\varepsilon + \frac{2\varepsilon}{\gamma}}$ 'holds in expectation'. Is it sufficient to guarantee that the solution provided by the algorithm is close to feasible with high probability?
* The continuous algorithm proposed in the paper theoretically outperforms the discrete algorithms, while the experiments only showcase the performance of the discrete algorithms. Are there any experimental results available for the continuous algorithm?

---

> ### Author Response · Authors · 2024-11-21
> **Author Rebuttal**
>
> > The motivation for considering FSC is unclear. The authors only say that FSM has been well studied while FSC has not. This can not be a good motivation
>
> > What are the motivations for FSC, e.g., real-world applications?
>
> We believe that FSC has broad applications within machine learning, and in the response to all reviewers and ACs we have included a detailed discussion on four different important applications of FSC. The applications include data summarization, data subset selection for training, neural network pruning, and influence in social networks. For more details, see https://openreview.net/forum?id=ULorFBST6X&noteId=vZ496FjX3r; a discussion of these applications has further been added to the paper in Appendix B.1 and is highlighted in blue.
>
> > The experiments demonstrate the advantages of the discrete algorithms on the FSC problem compared to the standard greedy algorithm; however, the solutions obtained by the discrete algorithms include many more elements. Moreover, the experiments lack the presentation of the effectiveness of the continuous algorithm.
>
> > The continuous algorithm proposed in the paper theoretically outperforms the discrete algorithms, while the experiments only showcase the performance of the discrete algorithms. Are there any experimental results available for the continuous algorithm?
>
> Continuous algorithms in submodular optimization can have relatively high query complexity because we do not necessarily have direct access to the multilinear extension, and instead have to make many noisy queries to $f$ and apply concentration bounds in order to approximate it (See Definition 3 in Section 1.2). For instance, with $\epsilon = 0.05$, $n = 5000$, and $\kappa = 10$, a single estimation of the multilinear extension value in Line 8 of the Decreasing-Threshold-Procedure (DTP) in Algorithm 3 requires approximately $5.33 \times 10^5$ queries. The total number of queries can reach up to $10^{12}$. In fact, many prior works on the multilinear extension have not included experiments due to the high computational cost. However, we agree with the reviewer that it would be interesting to know if the continuous algorithm can produce solutions of lower cardinality compared to our other algorithms, and so we have performed additional experiments evaluating our continuous algorithms heuristically on smaller problem instances.
>
> Given the high query complexity and the limited time available during the rebuttal period, we take 5 samples per estimation in Line 8 of the DTP and test the algorithms on a subgraph of the Twitch dataset with $n = 2000$ on the instance of maximum coverage. The results comparing different algorithms are included in the updated submission PDF (appendix, highlighted in blue). Notice that by taking fewer samples than is theoretically needed, this could potentially hurt the performance of the continuous algorithm. Despite this handicap, the cost of the continuous algorithm is smaller than the discrete algorithms under fairness constraint which includes the GREEDY-Fair and THRES-Fair algorithms. While the fair algorithms do still generally produce solutions of higher cost compared to the greedy without fairness constraint, this is to be expected since in order to be fair our algorithms have to select elements that are not optimal from the perspective of $f$ value.
>
> We have added these experiments into Appendix G of the updated version of our manuscript.

---

> > ### Comment · Reviewer_pFnE · 2024-11-21
> > **Reviewer Response**
> >
> > Thanks for the response. It addresses several questions of mine, including the motivation and the technical novelty. I raise my score to 6.

---

### Official Review · Reviewer_WbLT · 2024-11-03

**Soundness:** 3
**Presentation:** 3
**Contribution:** 3
**Rating:** 6
**Confidence:** 4

**Summary:**

This paper studies a variant of the classical submodular cover problem, in which, given a monotone submodular function, the goal is to pick an element set with the smallest size so that the value of this set is larger than a given threshold. This paper studies the fairness version of the problem, in which the ground element set is partitioned into k groups, and each group has an upper $u$ and lower bound $l$. A fair solution should pick at least $l$ elements and, at most, $u$ elements from each group. The goal is still to pick a fair solution with a minimum size such that the value of the picked set is larger than a given threshold.

The main contribution of this work is a bicriteria algorithm that achieves $\ln n$-approximation with a slightly violating fairness constraint. This is the essentially best possible since the classical submodular cover admits the set cover problem as a special case.

**Strengths:**

1. The studied problem is theoretically interesting and the presentation of this paper is clear. I appreciate that the authors give an intuitive description, which helps understand the algorithm's high-level idea.

2. The obtained results are almost tight for a bicriteria approximation. Based on my knowledge, these kinds of fairness constraints (upper and lower bounds for each group) always improve the difficulty of a problem a lot. It is reasonable to study the bicriteria approximation in this case.

3. The authors also include the experimental result for the set cover instance, demonstrating the proposed algorithms' efficiency.

**Weaknesses:**

1. The studied problem is theoretically interesting, but it is a little incremental. One needs to provide convincing motivation for the problem. It looks that the studied problem only adds an extra constraint to the classical submodular cover problem. The introduction section does not give a specific motivation for the studied problem.

2. There is no lower bound known in the paper. Namely, there is no lower bound for the bicriteria approximation. This may not be a big problem since these fair constraints are usually hard. But, one still needs to show how good is the trade-off obtained by the paper.

**Questions:**

I don't have any specific questions.

---

> ### Author Response · Authors · 2024-11-21
> **Author Rebuttal**
>
> > The studied problem is theoretically interesting, but it is a little incremental. One needs to provide convincing motivation for the problem. It looks that the studied problem only adds an extra constraint to the classical submodular cover problem. The introduction section does not give a specific motivation for the studied problem.
>
> We are confident that FSC has broad applications within machine learning, and in the response to all reviewers and ACs we have included a detailed discussion on four different important applications of FSC. The applications include data summarization, data subset selection for training, neural network pruning, and influence in social networks. For more details, see https://openreview.net/forum?id=ULorFBST6X&noteId=vZ496FjX3r; a discussion of these applications has further been added to the paper in Appendix B.1 and is highlighted in blue.
>
> It is true that FSC is the classical submodular cover problem with an additional constraint for fairness. However, we believe the problem is important both in terms of its broad range of applications and from a technical perspective. The maximization of a submodular function with a matroid constraint has received significant attention, but to the best of our knowledge, this work is the first to investigate a submodular cover problem with a matroid-type constraint. To tackle this problem, we introduced the novel concept of the $\beta$ matroid extension and proved an insightful result in Lemma 2, which establishes the relationship between a fairness matroid and its $\beta$-extension.
>
>
>
> > There is no lower bound known in the paper. Namely, there is no lower bound for the bicriteria approximation. This may not be a big problem since these fair constraints are usually hard. But, one still needs to show how good is the trade-off obtained by the paper.
>
> Because set cover is a special case of fair submodular cover (FSC), by the result of Feige (1998) it is not possible to achieve a  $(1-o(n)\ln(n))$ approximation to FSC unless NP has "slightly superpolynomial time algorithms". If we set $\epsilon=1/n$, our continuous algorithm (Algorithm 2) for FSM with the converting algorithm (Algorithm 1) achieves a $((1+\alpha)\ln(n)+1,1-7/n)$ bicriteria approximation guarantee for an instance of FSC, and so if $\alpha$ is close to 0, then the guarantee is very close to being feasible while having an approximation guarantee close to the lower bound. For other values of $\epsilon$, a lower bound has not yet been developed.

---

### Official Review · Reviewer_MRoR · 2024-11-04

**Soundness:** 4
**Presentation:** 3
**Contribution:** 3
**Rating:** 6
**Confidence:** 3

**Summary:**

The paper studies the fair submodular cover problem. In this problem, we are given a set of ground elements and a monotone submodular function defined over them. All ground elements are partitioned into $N$ groups, each group $c$ having a corresponding interval $[p_c,q_c]$. The objective is to select as few elements as possible such that the value of the submodular function is greater than or equal to a given threshold $\tau$, while ensuring that the proportion of selected elements from each group falls between $p_c$ and $q_c$.

The authors consider bicriteria approximation algorithms, where the threshold constraint may be violated by a factor. By establishing a relationship between fair submodular cover (FSC) and fair submodular maximization (FSM), they show that given a $(\gamma,\beta)$-approximation algorithm for FSM, they can convert it to an algorithm for FSC with an approximation ratio of $((1+\alpha)\beta, \gamma)$. if the given FSM algorithm is continuous, they can yield an approximation ratio of $((1+\alpha)\beta, ((1-3\epsilon)\gamma-2\epsilon)/(1+3\epsilon +2\epsilon/\gamma))$. Furthermore, they present nearly feasible algorithms that can admit $((1+\alpha)/\epsilon, 1-O(\epsilon))$-approximation and $((1+\alpha)(\ln(1/\epsilon)+1), 1-O(\epsilon))$. Finally, the proposed algorithms are investigated experimentally.

**Strengths:**

- The fair submodular cover problem makes sense. The paper makes theoretical contributions. A nice relationship between FSM and FSC is established in the paper and several bicriteria approximation algorithms are proposed. The high-level algorithmic ideas are clean.

- Both theoretical analysts and empirical evaluations are provided.

**Weaknesses:**

- The problem certainly holds theoretical interest. However, I am a bit concerned about whether it will engage the ICLR community. Could the authors provide some real-world applications?

- The authors did not compare the empirical running time of the proposed algorithms with the baselines. The theoretical running time looks horrible, so I wonder how long these algorithms will take in practice.

**Questions:**

See the weakness above.

---

> ### Author Response · Authors · 2024-11-21
> **Author Rebuttal**
>
> > The problem certainly holds theoretical interest. However, I am a bit concerned about whether it will engage the ICLR community. Could the authors provide some real-world applications?
>
> We are confident that FSC has broad applications within machine learning, and in the response to all reviewers and ACs we have included a detailed discussion on four different important applications of FSC. The applications include data summarization, data subset selection for training, neural network pruning, and influence in social networks. For more details, see https://openreview.net/forum?id=ULorFBST6X&noteId=vZ496FjX3r; a discussion of these applications has further been added to the paper in Appendix B.1 and is highlighted in blue.
>
> > The authors did not compare the empirical running time of the proposed algorithms with the baselines. The theoretical running time looks horrible, so I wonder how long these algorithms will take in practice.
>
> Generally, the runtime bottleneck for submodular optimization algorithms such as ours is the number of queries to the function $f$. Therefore, both our theoretical and empirical analyses focus on the number of queries to $f$, as is standard in submodular optimization. For GREEDY-Fair and THRES-Fair, the query complexities are $O\left(\frac{n|OPT|\log(|OPT|)}{\epsilon\log(\alpha+1)}\right)$ and $O\left(\frac{n\log(|OPT|)\log(|OPT|/\epsilon)}{\epsilon\log(\alpha+1)}\right)$, respectively. Compare these to the standard greedy algorithm that does not find a fair solution, which makes $O(n\ln(1/\epsilon)|OPT|)$ queries to $f$, so the theoretical runtime of our fair algorithms are similar to those of the greedy algorithm. In contrast, the continuous algorithm has a higher complexity of $O\left(\frac{n(1+\alpha)|OPT|\log^2\left(\frac{n}{\epsilon}\right)\log|OPT|}{\epsilon^4\alpha}\right)$. Our experiments aimed to demonstrate the relative performance of the algorithms rather than emphasize their absolute runtime efficiency, and comparing query complexity is most effective for this purpose. However, all experiments with each parameter configuration were able to run quickly and were completed within a 2-minute runtime.

---

> > ### Comment · Reviewer_MRoR · 2024-11-26
> >
> > Thank the authors for clarifying my concerns.

---

### Official Review · Reviewer_6Qua · 2024-11-08

**Soundness:** 4
**Presentation:** 3
**Contribution:** 3
**Rating:** 8
**Confidence:** 3

**Summary:**

This paper addresses the fair submodular cover problem, where a submodular function is defined over a set of elements that are divided into disjoint classes. The objective is to select a minimum number of elements such that the selected subset meets a target submodular value threshold and the number of selected elements of each class is within a predefined interval.

Although prior work has addressed fair submodular maximization, fair submodular cover has not been previously studied. The authors first provide a reduction from fair submodular maximization to fair submodular cover. It is worth noting that there was a reduction for the non-fair case, which does not directly apply here. Additionally, they develop new algorithms for fair submodular maximization, achieving improved results when applied within their reduction framework. For designing those algorithms, they leverage the result by El Halabi et al. (2020), which converts instances of fair submodular maximization into submodular maximization problems under matroid constraints.

**Strengths:**

The authors present the first known algorithm for fair submodular cover, which achieves a near-optimal approximation ratio close to that of the greedy algorithm for the non-fair version, with the added benefit of increased diversity in the selected elements. Also, the paper includes experimental results that support the practical effectiveness of their proposed methods and highlight the benefits of their approach in selecting diverse elements across classes.

**Weaknesses:**

The difference between the authors' reduction and the existing reduction from general submodular maximization to submodular cover could be elaborated on further. This distinction is somewhat unclear, and understanding the technical differences would enhance the clarity of their contribution.

The paper could benefit from a more detailed comparison between their fair submodular maximization algorithms and existing algorithms for submodular maximization under matroid constraints, especially for their best algorithm which is based on multilinear extension that already has been used for submodular maximization.

## Minor comment:
In lines 200 and 206, you use $U^c$ but I think $U_c$ is the correct one.

**Questions:**

- Could you clarify the technical differences between your reduction from fair submodular maximization to fair submodular cover and the standard reduction used in the non-fair setting?
- How do your algorithms for fair submodular maximization differ technically from previous approaches for submodular maximization under matroid constraints, particularly the one based on the multilinear extension?

---

> ### Author Response · Authors · 2024-11-21
> **Author Rebuttal**
>
> > The difference between the authors' reduction and the existing reduction from general submodular maximization to submodular cover could be elaborated on further.
>
> The standard reduction from an instance of submodular cover (SC) with objective $f$ and threshold constraint $\tau$ to an instance of submodular maximization (SM) with objective $f$ and budget $k$ involves iteratively doing the following procedure: A guess is made for the size of the optimal solution ($|OPT|$) to the instance of SC, and this guess as the budget along with $f$ are input into an algorithm for SM. The procedure is repeated with increasingly large guesses until a solution is found with $f$ value sufficiently close to $\tau$. This process is relatively straightforward since the two problems are dual to each other, and the conversion requires only a flip between the objective and the constraint. A clear description of the process is provided in (Iyer & Bilmes, 2013b).
>
> In contrast, in our fairness setting, the conversion from fair submodular cover (FSC) to fair submodular maximization (FSM) is less clear because of the more complex matroid structure of the fairness constraints. It is important to note that there is currently no existing formulation of SC that incorporates a matroid constraint, nor has any conversion process been developed to address such constraints. To address this, we devised a method where each guess of the size of the optimal solution for the instance of SC is used to construct an extended fairness matroid (we propose the concept of an extension to a matroid in Section 1.2). This matroid is then input as a constraint into a bicriteria FSM algorithm, such as those developed in Sections 3.1 and 3.2. Furthermore, post-processing (Lines 6–12 in Algorithm 1 and Lines 7-13 in Algorithm 4) is required for each guess to ensure the fairness constraint is met, unlike the non-fairness setting where no such post-processing is necessary. A final difference between our fair setting and the general setting is that we are the first to introduce a converting process for multilinear extension algorithms (Algorithm 4).
>
> We have included this discussion in Appendix B.2 of our updated version of the manuscript.
>
> > The paper could benefit from a more detailed comparison between their fair submodular maximization algorithms and existing algorithms ... submodular maximization.
>
> First, we note that the fair submodular maximization (FSM) problem can be converted into an instance of submodular maximization with a general matroid constraint, as shown by Lemma 3 in (El Halabi et al.,2020). Therefore, existing algorithms for SM with a general matroid constraint can also be applied to FSM. Such algorithms return a feasible solution to the instance of SM with a matroid constraint, and achieve an approximation guarantee of at best $1-1/e$. However, our goal with our algorithms for FSM is to use them as subroutines in the conversion process, and so we develop algorithms that find sets that are better approximations to the optimal solution than $1-1/e$ but are not necessarily feasible. i.e., we propose algorithms for FSM with bicriteria approximation ratio. To this end, we have developed the notion of an extension of the fairness matroid (see Definition 2 in Section 1.2) and shown that by transferring to this "bigger" constraint we can achieve better objective values. In the case of our continuous algorithm, this means traveling within an extended polytope of the $\beta$-extension of the fairness matroid. This approach enables the $f$ value of the solution set to approach arbitrarily close to the optimal objective.
>
> We have incorporated this discussion into Appendix B.3 of our updated version of the manuscript.

---

### Meta-Review · Area_Chair_cb4J · 2024-12-23

**Metareview:**

The paper studies a fair variant of the submodular cover problem. In the classical submodular cover problem, we are given a submodular function and a threshold value, and the goal is to select a subset of the items whose value exceeds the threshold. This work studies this problem with additional fairness constraints: the items are partitioned into disjoint groups and there is a lower bound and upper bound on the number of items that the solution needs to contain from each group. The paper gives a reduction from the fair submodular maximization problem to the fair submodular cover problem that converts bicriteria guarantees for the former problem to the latter problem. The paper also proposes several algorithms for the former problem that can be combined with the reduction to obtain bicriteria approximation guarantees for the fair submodular cover problem. The paper gives both a discrete and continuous algorithm, with the latter achieving improved approximation guarantees comparable to those known for the standard submodular cover problem but with a higher running time than the discrete algorithm.

This paper makes a valuable contribution to the literature on fair submodular maximization and it gives the first algorithms with provable bicriteria guarantees for the submodular cover problem. A reduction from the submodular maximization to the submodular cover problem was already known from prior work, but this reduction does not take into account the fairness constraints. The reviewers appreciated the theoretical contribution. The main concerns were regarding the motivation and applications for the fair submodular problem, and whether the contribution is of broad enough interest for the ICLR community. The author response and subsequent revision addressed these concerns sufficiently.

**Additional Comments On Reviewer Discussion:**

The authors' response clarified several of the reviewers questions and concerns, such as the questions regarding the novelty of the techniques and the potential applications of this work. The authors provided several examples of applications where an algorithm for fair submodular cover may be used.

---

### Decision · Program_Chairs · 2025-01-22

Accept (Poster)